# *Pogz* deficiency leads to transcription dysregulation and impaired cerebellar activity underlying autism-like behavior in mice

Reut Suliman-Lavie[1,9], Ben Title[2,9], Yahel Cohen[1], Nanako Hamada[3], Maayan Tal[1], Nitzan Tal[1], Galya Monderer-Rothkoff[1], Bjorg Gudmundsdottir[4], Kristbjorn O. Gudmundsson[5,6], Jonathan R. Keller[5,6], Guo-Jen Huang[7], Koh-ichi Nagata[3,8], Yosef Yarom[2✉] & Sagiv Shifman[1✉]

Several genes implicated in autism spectrum disorder (ASD) are chromatin regulators, including *POGZ*. The cellular and molecular mechanisms leading to ASD impaired social and cognitive behavior are unclear. Animal models are crucial for studying the effects of mutations on brain function and behavior as well as unveiling the underlying mechanisms. Here, we generate a brain specific conditional knockout mouse model deficient for *Pogz*, an ASD risk gene. We demonstrate that *Pogz* deficient mice show microcephaly, growth impairment, increased sociability, learning and motor deficits, mimicking several of the human symptoms. At the molecular level, luciferase reporter assay indicates that POGZ is a negative regulator of transcription. In accordance, in *Pogz* deficient mice we find a significant upregulation of gene expression, most notably in the cerebellum. Gene set enrichment analysis revealed that the transcriptional changes encompass genes and pathways disrupted in ASD, including neurogenesis and synaptic processes, underlying the observed behavioral phenotype in mice. Physiologically, Pogz deficiency is associated with a reduction in the firing frequency of simple and complex spikes and an increase in amplitude of the inhibitory synaptic input in cerebellar Purkinje cells. Our findings support a mechanism linking heterochromatin dysregulation to cerebellar circuit dysfunction and behavioral abnormalities in ASD.

[1] Department of Genetics, The Institute of Life Sciences, The Hebrew University of Jerusalem, Jerusalem, Israel. [2] Department of Neurobiology, The Institute of Life Sciences and Edmond & Lily Safra Center for Brain Sciences (ELSC), The Hebrew University of Jerusalem, Jerusalem, Israel. [3] Department of Molecular Neurobiology, Institute for Developmental Research, Aichi Developmental Disability Center, Kasugai, Japan. [4] Cellular and Molecular Therapeutics Branch, National Heart Lung and Blood Institutes (NHLBI)/National Institute of Diabetes and Digestive and Kidney Diseases (NIDDK), National Institutes of Health (NIH), Bethesda, MD, USA. [5] Mouse Cancer Genetics Program, Center for Cancer Research, National Cancer Institute at Frederick, Bldg. 560/12-70, 1050 Boyles Street, Frederick, MD 21702, USA. [6] Basic Research Program, Leidos Biomedical Research Inc, Frederick National Laboratory for Cancer Research, Bldg. 560/32-31D, 1050 Boyles Street, Frederick, MD 21702, USA. [7] Department and Graduate Institute of Biomedical Sciences, College of Medicine, Chang Gung University, Taoyuan, Taiwan. [8] Department of Neurochemistry, Nagoya University Graduate School of Medicine, Nagoya, Japan. [9] These authors contributed equally: Reut Suliman-Lavie, Ben Title. ✉email: yosef.yarom@mail.huji.ac.il; sagiv.shifman@mail.huji.ac.il

The etiology of autism spectrum disorder (ASD) has puzzled medical researchers for several decades, but in recent years there have been several breakthroughs. One of these is the increased appreciation regarding the importance of de novo mutations in ASD[1]. Exome sequencing studies revealed that hundreds of genes may contribute to ASD when disrupted by de novo mutations[2–5]. Furthermore, many of the genes implicated in ASD were also found to be associated with other types of neurodevelopmental disorders (NDDs)[6–10].

A major goal in ASD research is to identify common mechanisms that are shared by the numerous ASD-associated genes, aiming to explain why mutations in all these different genes commonly lead to ASD[11]. Systematic analyses of genes with de novo mutations in ASD showed enrichment for transcription regulators and chromatin modifiers[5,12], strongly suggesting that alteration in transcription and chromatin organization might have key pathogenic roles in ASD. Specifically, we noted that several of the high-confidence ASD risk genes are involved in heterochromatin formation, or participate in complexes that bind different isotypes of the human heterochromatin protein 1 (HP1) (e.g. *SUV420H1*[13], *ADNP*[14] and *POGZ*[15–19]). A key role of HP1 proteins is transcriptional silencing and modulation of chromatin architecture, including transcriptional repression of euchromatic genes[20]. Transcriptional repression is crucial for neurogenesis and for the proper function of post-mitotic neurons, whereas aberrant silencing can lead to NDDs[21,22]. To study the mechanisms by which mutations in chromatin-related genes contribute to ASD, we focused on the *POGZ* gene (POGO transposable element with ZNF domain). Not only is *POGZ* one of the most significantly associated genes with ASD, it is also consistently found to be a strong interactor with the three isotypes of HP1[23–25]. Expression analysis showed that POGZ is expressed in the developing mouse cortex and hippocampus in the early developmental stages, but decreases to lower expression in the nucleus of both cortical and hippocampal neurons at P30[26]. In the developing cerebellum, POGZ was detected dominantly in the nucleus of Purkinje cells, whereas in the granular and molecular layers, POGZ expression was observed at P15 and P30[26]. Recent studies provided compelling evidence that loss-of-function (LoF) mutations in *POGZ* are associated with abnormal development and behavior[15–19]. Many of the individuals with *POGZ* mutations, now known as White-Sutton syndrome [MIM: 616364], have developmental delay, and more than half are diagnosed with ASD. Individuals diagnosed with the disorder may exhibit intellectual disability, microcephaly, overly friendly behavior, short stature, hyperactivity and vision problems[16–18]. Despite the detailed phenotypes, the molecular, cellular and physiological mechanisms of this syndrome are still unclear.

To study the function of *POGZ*, we generated a mouse model with a heterozygous or homozygous nervous system-specific deletion of the *Pogz* gene. We found additive effects of *Pogz* on growth, development of the brain, behavior, gene expression and cerebellar physiology. Our study suggests that mutations in *Pogz* cause abnormal brain development and transcriptional dysregulation, which leads to neuronal and circuitry dysfunction, and eventually contributes to the behavioral and cognitive symptoms seen in humans with *POGZ* mutations.

## Results

### Nervous system-specific deletion of *Pogz* does not result in gross defects in brain anatomy. We studied the effect of *Pogz* dosage on the resulting phenotype by generating heterozygote and homozygote knockouts (KO) of *Pogz*. Since a complete KO of *Pogz* causes early embryonic lethality[27], we crossed conditional *Pogz* mice with transgenic Nestin$^{CRE}$ mice to produce heterozygous (cKO$^{+/−}$) and homozygous (cKO$^{−/−}$) mutations restricted to the central and peripheral nervous system, as well as control littermates without *Pogz* deletion (Fig. 1a).

The efficient deletion of POGZ in the brain was validated by immunostaining and western blots (Fig. 1b, c and Supplementary Fig. 1a). The immunostaining with POGZ antibodies also showed widespread expression of POGZ in neurons across the postnatal and adult brain of control mice (Fig. 1b and Supplementary Fig. 1a). To study whether *Pogz* knockout causes any anatomical abnormalities, we used Nissl staining, which revealed no gross brain anatomical defects in *Pogz*-deficient mice, but overall a smaller brain (Fig. 1d). Immunostaining with different neural markers in an early infant (P11) and adult mice did not show any detectable differences between *Pogz* cKO$^{−/−}$ and control mice in the cerebellum, hippocampus and cortex (Supplementary Fig. S1b–k), including the thickness of cortical layers (all $P > 0.05$) (Supplementary Fig. S2a–c). Furthermore, Golgi staining images revealed similar levels of dendritic spine density in the dentate gyrus of *Pogz* cKO$^{−/−}$ and control mice ($P = 0.98$) (Supplementary Fig. 2d, e).

### *Pogz*-deficient mice show growth delay, smaller absolute brain and defects in embryonic and adult neurogenesis. To test for association between *Pogz* and different mouse phenotypes, we tested for the additive (linear) relationship between the number of *Pogz* intact alleles (control = 2, cKO$^{+/−}$ = 1, and cKO$^{−/−}$ = 0) and the studied traits. In addition, we separately tested for phenotypic differences between pairs of genotypes. The most obvious phenotype associated with *Pogz* deficiency was body size (Fig. 2a). We observed a significant reduction in the body weight of *Pogz* cKO$^{−/−}$ relative to control littermates across development, with heterozygotes (*Pogz* cKO$^{+/−}$) showing an intermediate phenotype ($P = 1.1 × 10^{-12}$) (Fig. 2b). There was also a significant genotype by sex interaction for body weight, with mutant males more severely growth-delayed than females ($P = 0.016$). As some human individuals with *POGZ* mutations show microcephaly, we also measured total brain weight at P11, and found a significantly lower brain weight in *Pogz*-deficient mice ($P = 0.0042$) (Fig. 2c). However, brain weight relative to body weight was significantly larger in *Pogz* cKO$^{−/−}$ mice (Supplementary Fig. S2f), implying a larger impact on body weight. Given the absolute smaller brain, the known role of POGZ in mitosis[25] and the association of *POGZ* with microcephaly in humans, we hypothesized that *Pogz* may affect neurogenesis; therefore, we next studied how *Pogz* dosage influence embryonic and adult neurogenesis.

We studied the role of *Pogz* in embryonic neurogenesis using Pax6 and Tbr2 to mark apical progenitors and intermediate neural progenitors in the developing cerebral cortex (E15.5). We measured the size of the Pax6+ and Tbr2+ layers in comparable cerebral cortical sections from control, *Pogz* cKO$^{+/−}$ and *Pogz* cKO$^{−/−}$ littermates and found that the length of the Tbr2+ layer was modestly increased in the *Pogz*-deficient mice ($P = 0.042$; 7.3% and 10.8% increase in *Pogz* cKO$^{+/−}$ and *Pogz* cKO$^{−/−}$, respectively), whereas no significant difference was detected for the Pax6+ layer (Fig. 2d–f). Additionally, there was no significant difference in the thickness of the entire cortex or the cortical plate (CP) (Supplementary Fig. S3a–c). To determine whether changes in mitosis can contribute to the observed increase in Tbr2+ layer, we stained the sections for phospho-histone H3 (pHH3), which marks mitotic cells with condensed chromosomes. The number of pHH3+ cells was significantly reduced in *Pogz*-deficient mice ($P = 4.5 × 10^{-5}$; 19.0% and 43.8% decrease in *Pogz* cKO$^{+/−}$ and *Pogz* cKO$^{−/−}$, respectively) (Fig. 2g, h). Next, we examined cell-cycle progression using in utero electroporation of two different shRNA against *Pogz*. The efficacy of the inhibition was assessed

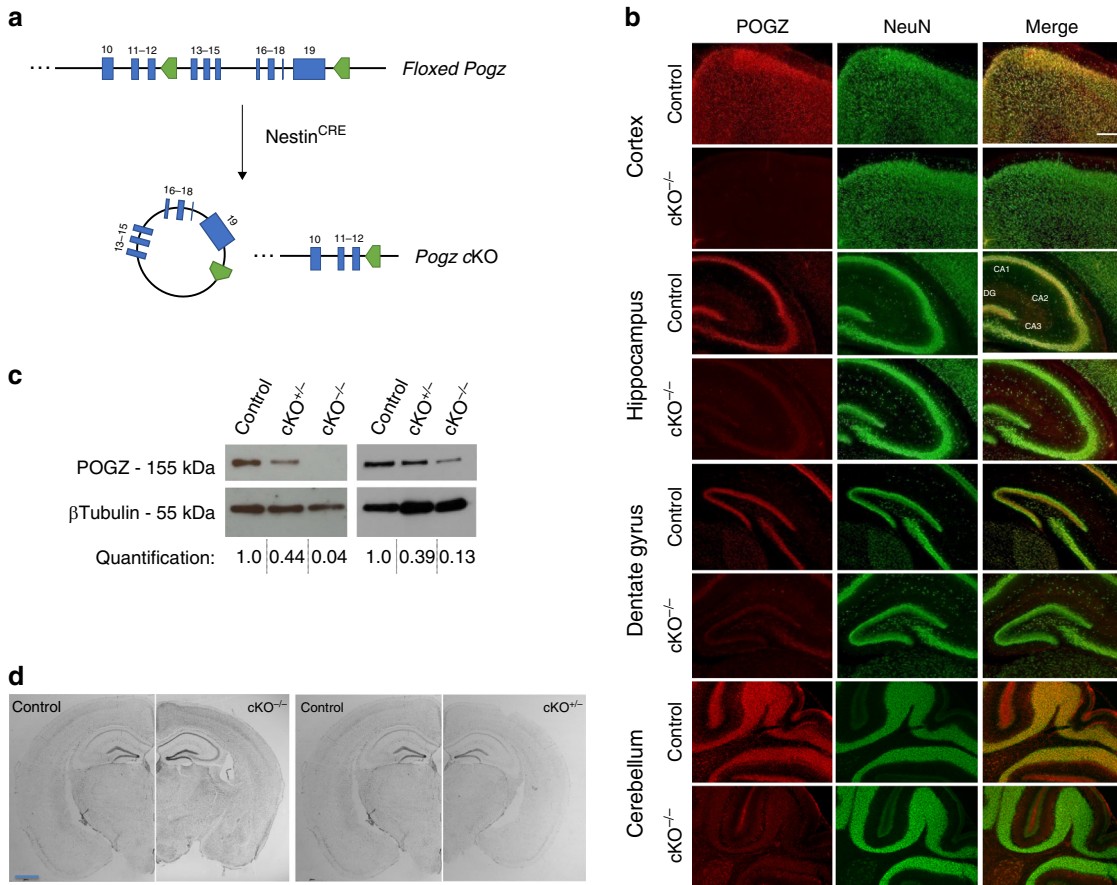

**Fig. 1 *Pogz* deficiency mouse model does not exhibit gross defects in brain anatomy. a** Schematic overview of *Pogz* knockout strategy. *Pogz* gene exons 13−19 (blue) were bounded by *loxP* sites (green). *Pogz* cKO$^{+/-}$ and *Pogz* cKO$^{-/-}$ mice were generated with a mutation restricted to the nervous system, by crossing floxed *Pogz* mice with *Nestin* Cre mice. **b** Immunofluorescence staining of *Pogz* cKO$^{-/-}$ and control mice brains (P11) using antibodies against POGZ (red) and NeuN (green). $n = 3$ control, 3 cKO$^{-/-}$ in one independent experiment. Scale bar = 100 μm. **c** Western blots analysis of whole-cell lysates extracted from E14.5 mice cortices. $n = 3$ control, 3 cKO$^{+/-}$, 3 cKO, in three independent experiments. **d** Nissl staining of coronal sections from *Pogz* cKO$^{-/-}$ and cKO$^{+/-}$ beside sections from control mice. $n = 6$ control, 6 cKO$^{+/-}$, 3 cKO$^{-/-}$ in two independent experiments. Scale bar = 1 mm.

by western blotting and immunofluorescence (Supplementary Fig. S3d, e). We studied cell-cycle exit by electroporation of GFP vector together with the *Pogz* shRNAs or control vector at E14.5 and 24 h after the electroporation the pregnant mice were injected with 5-ethynyl-2′-deoxyuridine (EdU) to label cells in the S phase. After 24 h the electroporated brains of the embryos (E16.5) were stained for EdU, Ki67 and GFP (Fig. 2i). We measured the ratio between cells still in cell cycle (EdU+/Ki67+/GFP+) and cells that exited the cell cycle (GFP+/EdU+/Ki67−). This ratio was significantly decreased in cells with *Pogz* shRNA compared with control shRNA ($P = 0.024$; 28.5% decrease) (Fig. 2j), indicating that cell-cycle exit was accelerated by *Pogz* knockout. These results show that *Pogz* is important for proper cell-cycle progression in the developing mouse cortex.

We next studied adult neurogenesis in the dentate gyrus (Fig. 2k and Supplementary Fig. S3f). We found a decrease in the number of cells labeled by the immature neurons marker, doublecortin (DCX), as a function of *Pogz*-deficiency levels ($P = 4.0 \times 10^{-5}$; 17.4% and 30.6% decrease in *Pogz* cKO$^{+/-}$ and *Pogz* cKO$^{-/-}$, respectively) (Fig. 2k). Lower levels of immature neurons can be caused by decreased proliferation and/or reduced cell survival. Proliferation was not significantly associated with *Pogz*-deficiency levels, as indicated by the number of cells labeled by the Ki67 marker ($P = 0.18$) (Fig. 2l). In contrast, the number of BrdU+ cells 21 days post-BrdU injection was significantly

reduced in *Pogz*-deficient mice ($P = 0.0027$; 28.4% and 34.3% decrease in *Pogz* cKO$^{+/-}$ and *Pogz* cKO$^{-/-}$, respectively) (Fig. 2m). Thus, our findings imply that adult neurogenesis is decreased in *Pogz*-deficient mice due to reduced cell survival.

**_Pogz_-deficient mice exhibit abnormal motor and social behavior.** One of the major phenotypes of White−Sutton syndrome is delayed psychomotor development[15,19]. In addition, autistic features are common, with some individuals showing overly friendly behavior. A summary of the tests performed relevant to the human phenotypes and the results obtained are presented in Supplementary Table S1.

To test motor coordination and strength we used the accelerating rotarod and horizontal bar tests. In the rotarod assay, we found differences in motor learning between genotypes, as indicated by the significant test for genotype by trial interaction ($P = 0.023$) (Fig. 3a). Motor coordination deficits were also evident in the horizontal bar test, where the degree of *Pogz* deficiency was significantly associated with reduced scores ($P = 0.0022$) (Fig. 3b). A significantly lower score was found for *Pogz* cKO$^{-/-}$ compared to *Pogz* cKO$^{+/-}$ ($P = 0.0028$) and control littermates ($P = 0.0010$). Motor deficits can affect locomotor activity, but the distance moved in the open field was not significantly altered by *Pogz* deficiency levels ($P = 0.26$) (Supplementary Fig. S4a).

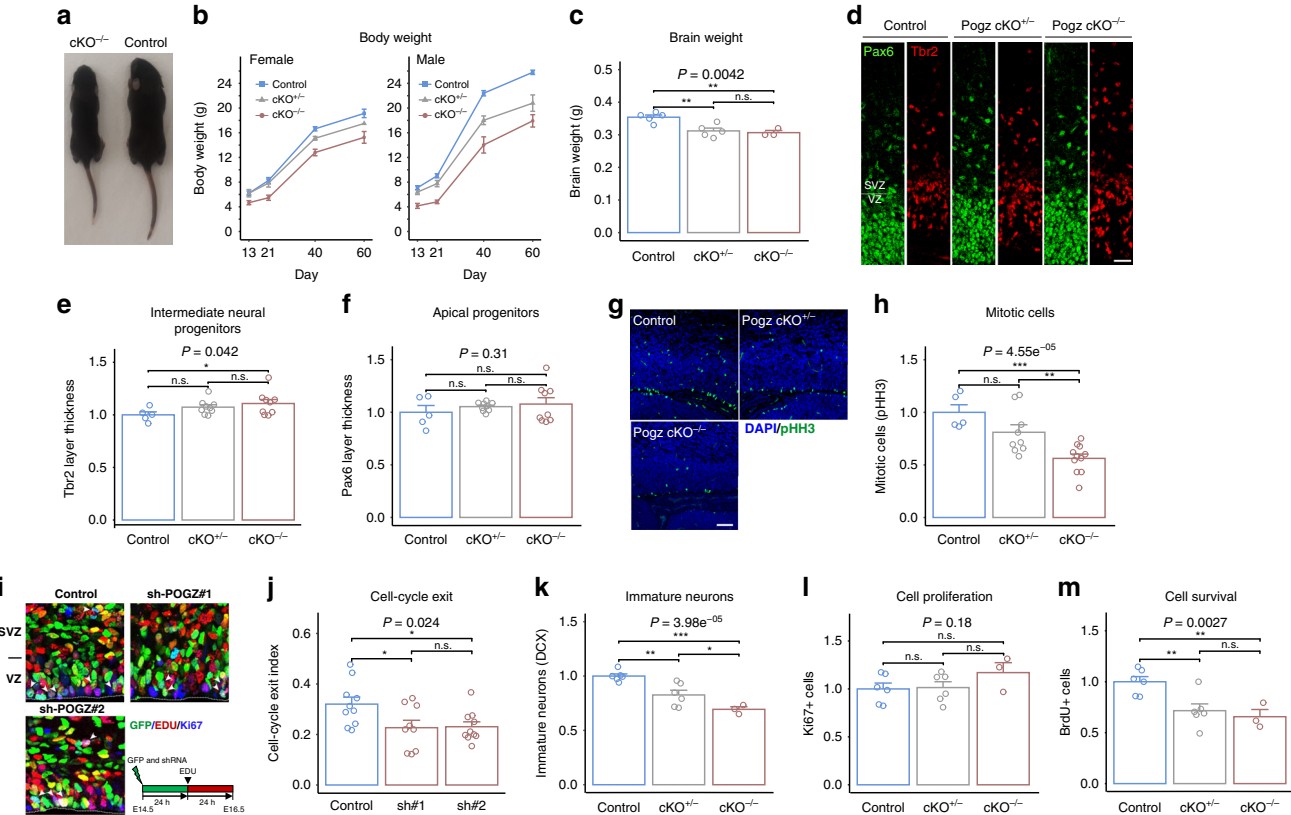

**Fig. 2 *Pogz*-deficient mice show growth delay, changes in brain size, and altered neurogenesis. a** Comparison of the body size of *Pogz* cKO$^{-/-}$ and control mice littermates (P11). **b** Body weight in four time points (P13, P21, P40, P60). Repeated measures ANOVA; $n = 16$ control (blue), 7 cKO$^{+/-}$ (gray), 7 cKO$^{-/-}$ (red), $P = 1.1 \times 10^{-15}$. **c** Brain weight at P11. $n = 5$ control, 5 cKO$^{+/-}$, and 3 cKO$^{-/-}$. **d** Coronal sections of embryonic cerebral cortex (E15.5) stained with anti-Pax6 and Tbr2 antibodies. Scale bar $= 50\,\mu$m. VZ ventricular zone, SVZ subventricular zone. **e, f** Quantification of the thickness of the Tbr2+ and Pax6+ layers. Values are proportion relative to control mice. $n = 5$ control, 9 cKO$^{+/-}$, 9 cKO$^{-/-}$ in four independent experiments. **g, h** Coronal sections of cerebral cortices (E15.5) stained with DAPI and antibody against phosphorylated histone 3 (pHH3). Scale bar $= 100\,\mu$m. **h** Quantification of the number of the pHH3+ cells. Values are proportion relative to control mice. $n = 5$ control, 9 cKO$^{+/-}$, 11 cKO$^{-/-}$ in two independent experiments. **i, j** Effects of Pogz-silencing on cell-cycle exit. **i** Differentiated neurons are EdU/GFP-double-positive while EdU/Ki67/GFP-triple-positive cells maintain progenitor potency. Arrowheads indicate triple-positive cells. Scale bar $= 10\,\mu$m. At the bottom right is the scheme of the cell-cycle exit assay. **j** Quantification of EdU/Ki67/GFP-triple-positive cells among GFP/EdU double-positive ones (Cell exit index). Two sections were used per brain (more than 1600 cells were analyzed in each condition). $n = 10$ control, 9 sh-Pogz#1 (sh#1), 10 sh-Pogz#2 (sh#2) from two in utero independent experiments. **k−m** Reduced adult neurogenesis associated with *Pogz* deficiency. Quantification of either **k** DCX-positive cells, **l** Ki67-positive cells, **m** BrdU-positive cells in the dentate gyrus, 21 days post injection. Numbers are proportion relative to control mice. $n = 6$ control, 6 cKO$^{+/-}$, 3 cKO$^{-/-}$. The P values at the top of the plots are for the association between the quantitative measurements and the number of intact *Pogz* alleles calculated with a linear regression model. Pairwise comparisons between genotypes was calculated by two-tailed *t* test and the significance is represented by: *$P < 0.05$; **$P < 0.01$; ***$P < 0.001$; n.s. not significant. Quantitative data are mean ± SEM.

We found a significantly lower performance of *Pogz*-deficient mice in Morris water maze and T-maze suggesting deficits in spatial learning, spatial memory and working memory. In the Morris water maze, *Pogz* deficiency was associated with lower spatial learning ability based on distance and latency to reach the hidden platform, ($P = 4 \times 10^{-5}$; $P = 2 \times 10^{-10}$, respectively) (Fig. 3c, d), and lower memory based on the probe test, when the hidden platform was removed ($P = 0.0091$) (Supplementary Fig. S4b). In the T-maze spontaneous alternation test, there was a small but significant decrease in the percentage of alternations in *Pogz*-deficient mice ($P = 0.036$) (Fig. 3e), implying deficits in working memory.

To assess anxiety-related responses, we performed the elevated plus maze and observed no significant differences in the time spent in the open arms ($P = 0.76$) (Supplementary Fig. S4c). Similarly, there was no significant difference in the duration of time spent in the center of the open field ($P = 0.76$) (Supplementary Fig. S4d). Next, we used the marble burying test to assess

repetitive digging, which is also known to be influenced by antidepressants[28,29], and self-grooming as another measure of repetitive behavior. Marble burying was significantly reduced in *Pogz*-deficient mice ($P = 0.00044$) (Fig. 3f), while grooming was not significantly different ($P = 0.21$) (Fig. 3g).

We next analyzed social behavior in a three-chamber social approach task. Consistent with the overly friendly phenotype seen in some human individuals with *POGZ* mutations, *Pogz* deficiency levels were significantly associated with increased duration near the cup and in the chamber of a stranger mouse ($P_{Cup} = 0.0071$, $P_{Chamber} = 0.0029$) (Fig. 3h and Supplementary Fig. S4e). In the social novelty test, there was no significant difference between genotypes ($P > 0.3$ and Supplementary Fig. S4f, g). To further study social behavior, we examined social interaction between two freely moving mice in the reciprocal social interaction assay. *Pogz*-deficient mice showed a nonsignificant increase in total time of social interaction ($P = 0.091$) (Fig. 3i), with no significant increase in nose-to-nose sniffing

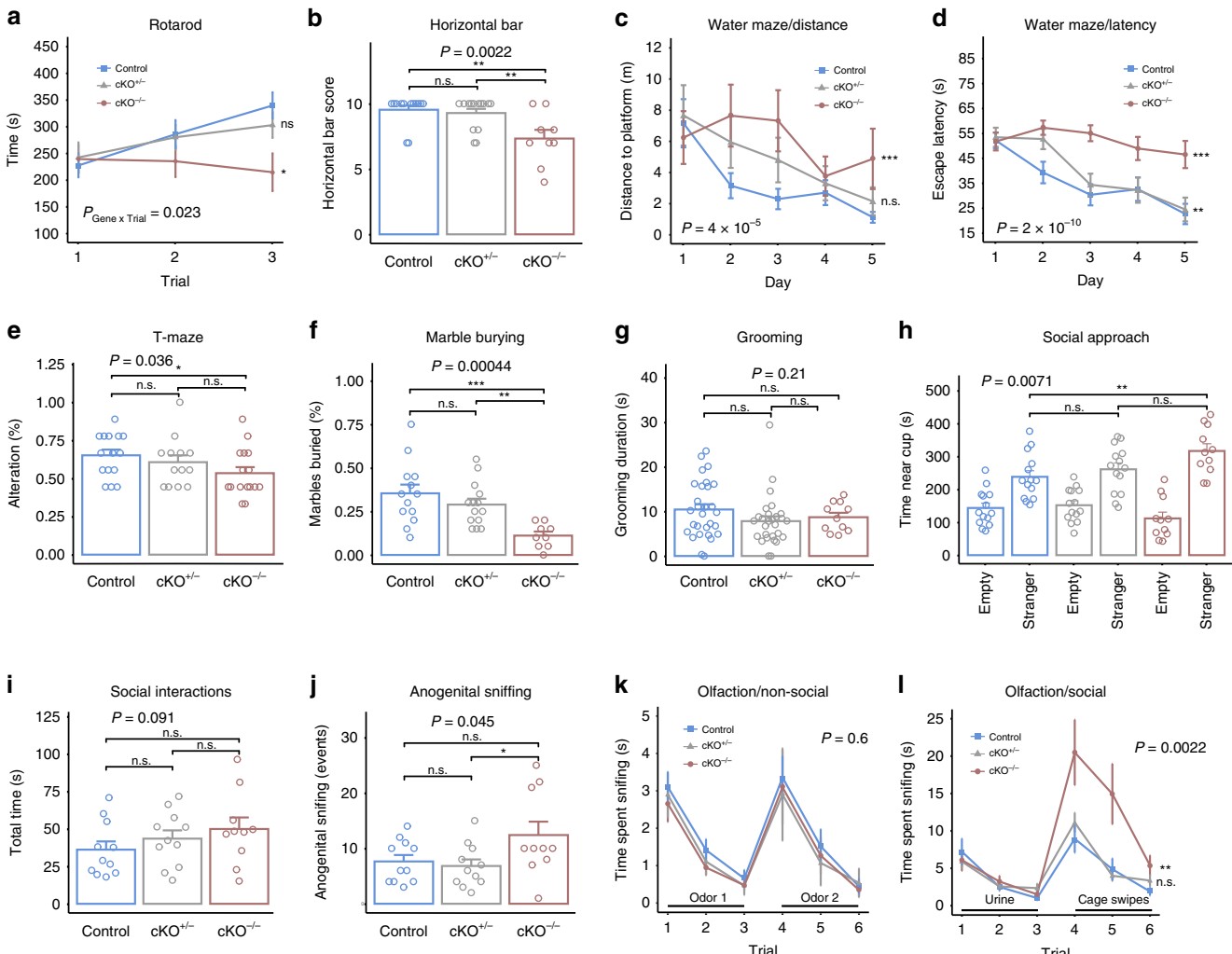

**Fig. 3 *Pogz*-deficient mice show abnormal motor, cognitive and social behavior. a** Time spent on the accelerated rotarod for three consecutive trials. Repeated measures ANOVA; $n = 26$ control (blue), 25 cKO$^{+/-}$ (gray), 11 cKO$^{-/-}$ (red). **b** Time taken to cross or fall from the horizontal bar calculated as a score. $n = 13$ control, 14 cKO$^{+/-}$, 9 cKO$^{-/-}$. **c, d** Distance and latency to reach the platform in the Morris water maze. Repeated measures ANOVA; $n = 7$ control, 5 cKO$^{+/-}$, 5 cKO$^{-/-}$. **e** Percentage of correct responses in the T-maze forced alternation task across 10 trials. $n = 15$ control, 13 cKO$^{+/-}$, 15 cKO$^{-/-}$. **f** Percentage of marbles buried in the marble burying test. $n = 13$ control, 14 cKO$^{+/-}$, 9 cKO$^{-/-}$. **g** Time spent self-grooming during the open field. $n = 30$ control, 28 cKO$^{+/-}$, 11 cKO$^{-/-}$. **h** Time spent near the cup with a stranger or the empty cup as measured in the three-chamber sociability test. $n = 14$ control, 14 cKO$^{+/-}$, 11 cKO$^{-/-}$. **i** Total time of social interaction events in pairs of same sex mice freely behaving in the cage. $n = 11$ control, 11 cKO$^{+/-}$, 10 cKO$^{-/-}$. **j** Number of nose-to-anogenital sniffing events in pairs of same sex mice freely behaving in the cage. $n = 11$ control, 11 cKO$^{+/-}$, 10 cKO$^{-/-}$. **k** The time spent sniffing nonsocial odors in the olfactory habituation–dishabituation test. Repeated measures ANOVA; $n = 11$ control, 12 cKO$^{+/-}$, 12 cKO$^{-/-}$. **l** Time spent sniffing urine and cage odors in the olfactory habituation–dishabituation test. Repeated measures ANOVA; $n = 20$ control, 23 cKO$^{+/-}$, 9 cKO$^{-/-}$. The P values at the top of the plots (**b**−**i**) are for association between the quantitative measurements and the number of intact *Pogz* alleles calculated with a linear regression model (unless stated otherwise). Pairwise comparisons between genotypes was calculated by two-tailed *t* test and the significance is represented by: *$P < 0.05$; **$P < 0.01$; ***$P < 0.001$; n.s. not significant. For repeated measures, the symbols represent the significance of the difference from the control. Quantitative data are mean ± SEM.

($P = 0.13$) (Supplementary Fig. S4h), whereas nose-to-anogenital sniffing was significantly increased ($P = 0.045$) (Fig. 3j). To test if *Pogz*-deficient mice can detect and differentiate between odors, we tested the mice in an olfactory habituation−dishabituation task with nonsocial and social odors. In the nonsocial odors, there was no significant effect of *Pogz* dosage ($P = 0.60$) (Fig. 3k). For the social odors, we used urine and cage swipes (from age- and sex-matched mice). In the social odors, all genotypes showed habituation to the odor and dishabituation to the novel odor, but there was a significant difference in sniffing time. Consistent with the increased social interest of *Pogz*-deficient mice, the sniffing time of cage swipes was significantly increased in *Pogz* cKO$^{-/-}$

(genotype effect, $P = 0.0022$; genotype by trial interaction, $P = 0.0087$) (Fig. 3l).

Our analysis supports the association of *Pogz* levels with multiple behaviors using an additive genetic model. However, when comparing with controls most behavioral tests showed significant differences only for *Pogz* cKO$^{-/-}$ and a smaller nonsignificant effect for *Pogz* cKO$^{+/-}$. To evaluate the fit of the additive genetic model, we compared the Akaike information criterion (AIC) for three possible genetic models: recessive, additive and dominant models with behavioral results that do not include repeated measures. The result of the analysis showed that none of the six behaviors are best explained by the dominant

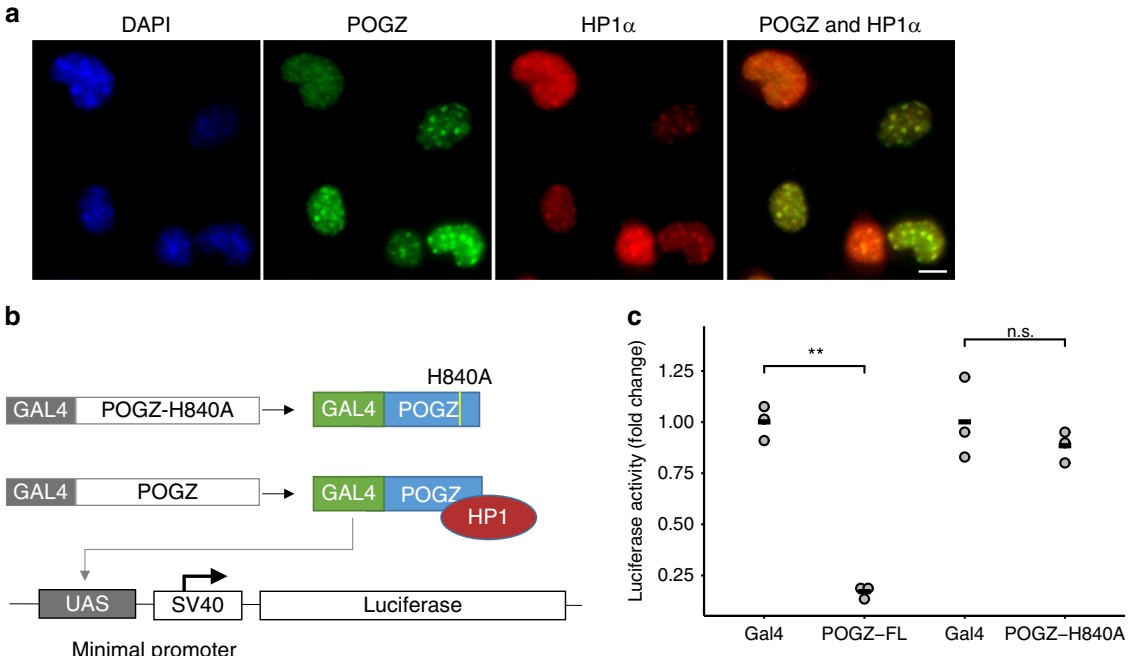

**Fig. 4 POGZ localizes to pericentric chromatin and represses transcription depending on HPZ. a** Immunofluorescence staining of mouse neuronal cells (N2A) using antibodies against POGZ (green), HP1α (red), and DAPI (blue). Three replicates in one experiment. **b** Schematic representation (not to scale) of the *GAL4*-fusion proteins and a reporter gene containing GAL4-binding sites (UAS) located upstream of the SV40 promoter and luciferase gene (UAS-SV40-luc). The H840A mutation, that change the conserved amino acid Histidine to Alanine in the HPZ motif, was previously shown to abolish the interaction between POGZ and HP1α[25]. **c** Fold change of luciferase activity in HEK293 cells expressing: Gal4, Gal4 DNA binding domain (DBD) fused to full-length POGZ (POGZ -FL), or Gal4-DBD fused to POGZ with a mutation in the HPZ sequence (POGZ H840A). Three independent experiments for each condition. Two-tailed *t* test; **$P < 0.01$; n.s. not significant. The black horizontal line indicates the mean.

model, two behaviors (horizontal bar and anogenital sniffing) are best explained by a recessive model (effect only in cKO$^{-/-}$), and four out of six are best explained by an additive model (intermediate effect in cKO$^{+/-}$ and larger effect in cKO$^{-/-}$) (Supplementary Table S2).

**POGZ represses transcription depending on the HP1-binding zinc finger-like domain.** Since POGZ was found to be an integral part of the HP1 protein complexes[23–25], we hypothesized that POGZ plays a key role in transcription regulation. Based on immunostaining, POGZ is distributed at pericentric heterochromatin, in a similar way to HP1α (Fig. 4a). To assess whether POGZ regulates transcription, we performed a dual-luciferase reporter assay in HEK293 cells (Fig. 4b). We transiently introduced a GAL4 DNA-binding domain (GAL4-DBD) fused with full-length POGZ (POGZ-FL) to the cells and noticed that transcription of the reporter gene was significantly inhibited by POGZ-FL ($P = 0.0012$) (Fig. 4c). POGZ interacts with HP1 proteins through an HP1-binding zinc finger-like (HPZ) domain, rather than the canonical PxVxL motif[25]. It was previously shown that a single mutation in any Cys or His residues in the HPZ motif, including an H840A mutation, abolishes the interaction with HP1α[25]. To test if the interaction with HP1 is essential for transcription repression, we used POGZ with H840A mutation in the HPZ domain (Fig. 4b). POGZ-H840A failed to repress reporter gene transcription ($P = 0.42$) (Fig. 4c), suggesting that POGZ functions as a negative regulator of transcription depending on the interaction with HP1 proteins.

**Transcriptional dysregulation in the brain of *Pogz*-deficient mice.** Given that POGZ is involved in transcriptional repression, we wanted to study how POGZ affects gene expression in the brain. While *Pogz* has widespread expression in the mouse brain, it is highly expressed in the cerebellum in both human and mouse (Supplementary Fig. S5a, b). We therefore focused on gene expression in the cerebellum of adult mice and compared it to the hippocampus, where *Pogz* is less expressed. We performed the differential expression analysis using RNA-seq of six RNA samples from each genotype, *Pogz* cKO$^{-/-}$, *Pogz* cKO$^{+/-}$ and control littermates, including males and females. To identify differential expression that is associated with *Pogz* dosage, we used a statistical model that accounted for sex and treated the *Pogz* genotypes as a quantitative measure (2, 1, and 0) across genes that were robustly expressed (counts per million (cpm) > 1 in at least a third of the samples; hippocampus: $n = 14,339$, cerebellum: $n = 14,027$) (Table S3). At significance cutoffs corresponding to false discovery rate (FDR) < 0.05, we found 636 genes that were differentially expressed in the hippocampus, among them 46 with fold change > 1.5 and 15 with fold change > 2 (fold change is the average effect of allele substitution) (Fig. 5a–c). In the cerebellum, we identified a threefold increase in the number of differentially expressed genes: 1916 at significance cutoffs corresponding to FDR < 0.05, 196 with fold change > 1.5 and 52 with fold change > 2 (Fig. 5d–f). Most differentially expressed genes showed an additive effect, with the expression level in cKO$^{+/-}$ in between the levels of cKO$^{-/-}$ and control (Fig. 5c, f and Supplementary Fig. S5c, d). Consistent with the role of POGZ in gene repression, the majority of differentially expressed genes with absolute fold change > 1.5 (hippocampus: 67%, $P = 0.026$; cerebellum: 77%, $P = 4.6 \times 10^{-14}$) or 2.0 (hippocampus, 93%, $P = 0.00098$; cerebellum: 83%, $P = 2.0 \times 10^{-6}$) were upregulated. There was no significant difference in the proportions when genes with absolute fold change below 1.5 were included (hippocampus: 47%, $P = 0.14$; cerebellum, 52%, $P = 0.079$) (Fig. 5a, d).

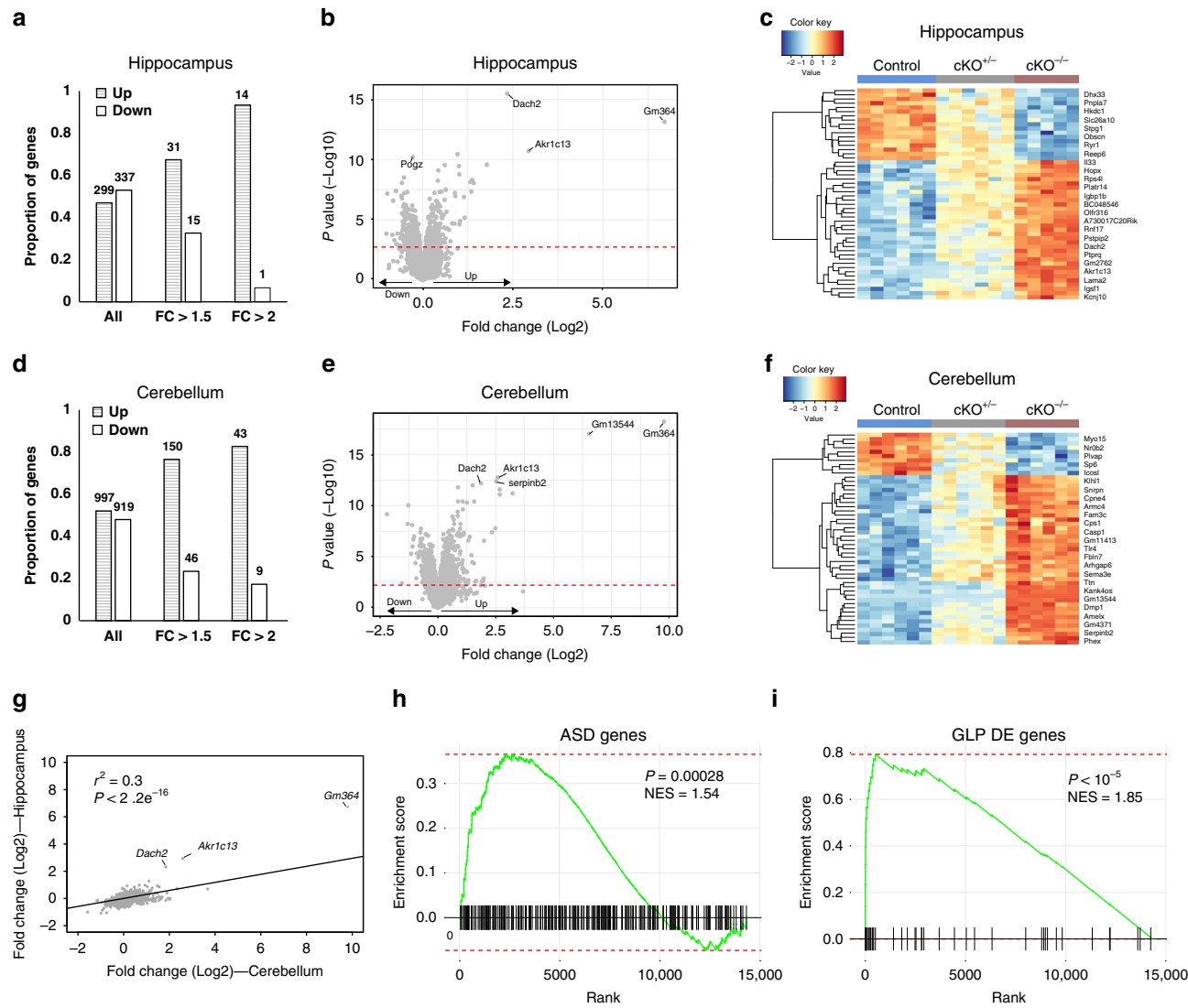

**Fig. 5 *Pogz* deficiency leads to transcriptional dysregulation. a, d** Proportion of up- and downregulated genes in the **a** hippocampus or **d** cerebellum of *Pogz*-deficient mice. Bars correspond to all genes (left), genes with fold change > 1.5 (middle) and genes with fold change > 2 (right). **b, e** Volcano plot showing differentially expressed genes in the **b** hippocampus or **e** cerebellum. *P* values were calculated by edgeR. The red line indicates FDR = 0.05. Up and down (bottom left) indicate genes upregulated and downregulated (respectively) in the *Pogz*-deficient mice. **c, f** Heatmap of the 50 most significant genes in the **c** hippocampus or **f** cerebellum. The color corresponds to the scaled expression levels (blue—negative, red—positive). **g** Correlation between fold changes of gene expression in the hippocampus and the cerebellum. The correlation, significance and linear regression were calculated using an iteratively re-weighted least squares algorithm (lmrob function in the 'Robustbase Package' in R). **h, i** GSEA enrichment plots of genes differentially expressed in the hippocampus of *Pogz*-deficient mice (see also Supplementary Table S3). The significance of the enrichment was calculated using fgseaMultilevel function in the 'fgsea Package' in R. **h** ASD candidate genes (SFARI gene base, score 1−3) are enriched in gene upregulated in the hippocampus. **i** Genes differentially expressed in the GLP mouse model are significantly enriched among genes differentially expressed in the *Pogz* mouse model.

Despite the differences in the number of differentially expressed genes between the hippocampus and cerebellum, the changes in expression were related. First, 41.1% of genes differentially expressed in the hippocampus were also differentially expressed in the cerebellum ($P < 2.2 \times 10^{-16}$). Second, the fold change across all commonly expressed genes was significantly correlated between the hippocampus and cerebellum ($r^2 = 0.30$, $P < 2.2 \times 10^{-16}$), but the slope of the regression line indicated that gene expression changes in the hippocampus are consistently lower compared to the cerebellum (Fig. 5g). Three genes were upregulated in the hippocampus more than expected based on the fold change in the cerebellum: *Dach2*, *Akr1c13*, and *Gm364*. *Akr1c13* is a rodent-specific gene expressed exclusively in

the stomach, liver, and ileum[30]; *Gm364*, which showed upregulation only in cKO$^{-/-}$ (Supplementary Fig. 5c, d), is a mouse-specific gene, highly expressed in the adult mouse testis. *Dach2 is* a transcription factor[31] that is the most significantly differentially expressed gene in the hippocampus (adjusted $P = 4.5 \times 10^{-12}$), and one of the top significant genes in the cerebellum (adjusted $P = 4.9 \times 10^{-9}$). Relative to control littermates, *Dach2* was upregulated in the hippocampus of *Pogz* cKO$^{-/-}$ mice by 24.3 and by 4.5-fold in *Pogz* cKO$^{+/-}$ (Supplementary Fig. S5e). In the cerebellum, *Dach2* was upregulated by 12.5-fold in *Pogz* cKO$^{-/-}$ mice and by 3.8-fold in *Pogz* cKO$^{+/-}$ mice (Supplementary Fig. S5f). We confirmed the upregulation of *Dach2* by quantitative PCR (qPCR) in the

hippocampus, cerebellum, and the cortex (Supplementary Fig. S5g). Based on reads distribution from the RNA-seq, an uncharacterized *Dach2* transcript that starts from the middle of exon 8 is upregulated in the *Pogz*-deficient mice (Supplementary Fig. S5h, i)—a short transcript which is observed uniquely in neurons (Supplementary Fig. S5j).

To identify processes affected by *Pogz* deficiency, we analyzed the differentially expressed genes for enrichment of gene ontology using a gene set enrichment analyses (GSEA). The upregulated genes (FDR < 0.05) in the cerebellum were enriched for GO terms related to organ development, particularly of the nervous system (e.g. central nervous system development and neurogenesis), metabolic processes, myelin sheath, synapse, and sex differentiation (Supplementary Table S4). The downregulated genes in the cerebellum were enriched for DNA repair, RNA processing, and chromosome organization. In the hippocampus, the upregulated genes were also enriched for genes involved in metabolic processes (e.g. oxidoreductase activity), myelin sheath, and sex differentiation. The downregulated genes in the hippocampus were enriched for terms related to synapses such as excitatory synapse and calcium channel complex. We also tested the enrichment of ASD candidate genes (SFARI gene score 1, 2 or 3) among the differentially expressed genes. There was a significant enrichment of ASD genes among genes downregulated in the hippocampus ($P = 2.8 \times 10^{-4}$) (Fig. 5h), but not for the cerebellum ($P = 0.75$).

To investigate whether changes in gene expression might underlie the observed mouse phenotypes, we tested for enrichment of MGI mammalian phenotypes associated with the differentially expressed genes. We found that genes upregulated in the cerebellum were most significantly enriched with decreased body size, respiratory distress, perinatal lethality, abnormal skeleton morphology, and abnormal nervous system electrophysiology (Supplementary Table S4). Differentially expressed genes in the hippocampus did not show any significant enrichment for mammalian phenotypes (FDR < 0.05).

We next wanted to compare our differential expression findings to genes known to be regulated by proteins interacting with POGZ. We were able to identify only one other study that examined changes in gene expression in the mouse brain for two interactors of POGZ, GLP and G9a (EHMT1/EHMT2). In that study, conditional knockout of GLP/G9a in forebrain postnatal neurons resulted in derepression of non-neuronal genes in multiple brain regions, including the hippocampus (the cerebellum was not tested)[32]. *G9a* and *GLP* are two genes in the histone methyltransferase complex that interact with HP1 and POGZ, they are responsible for mono- and di-methylation of H3K9, and are required for transcription repression at euchromatic and facultative heterochromatin[33–35]. To test if *Pogz* controls a similar set of genes we compared the differential gene expression in the hippocampus between the two studies using GSEA. Remarkably, we found a highly significant enrichment of the differentially expressed genes in GLP cKO (Fig. 5i) and G9a cKO (Supplementary Fig. S5k) with the differentially expressed genes in *Pogz*-deficient mice ($P < 10^{-5}$). Among the overlapping genes were two of the upregulated genes discussed above, *Dach2* and *Akr1c13*.

**Pogz-deficient mice show a reduction in spontaneous firing rate of Purkinje cells in both simple and complex spikes.** As the cerebellum is associated with the motor and transcriptional abnormalities of *Pogz*-deficient mice, we examined cerebellar activity in anesthetized animals, focusing on Purkinje cells (PCs) activity. We performed extracellular recordings of PCs in lobules V−VI (examples of unit recording are shown in Fig. 6a). Complex spikes (CS) were readily identified and frequencies of

spontaneous simple spike (SS) and CS were calculated. We observed a reduction in spontaneous SS frequency as a function of *Pogz* deficiency levels ($P = 0.00012$; Control: $45.0 \pm 28.8$ Hz; cKO$^{+/−}$: $32.5 \pm 18.0$ Hz; cKO$^{−/−}$: $20.4 \pm 16.0$ Hz) (Fig. 6b). Similarly, a significant reduction in spontaneous CS frequency was also associated with *Pogz* deficiency levels ($P = 0.00030$; Control: $0.87 \pm 0.60$ Hz; cKO$^{+/−}$: $0.80 \pm 0.48$ Hz; cKO$^{−/−}$: $0.37 \pm 0.22$ Hz) (Fig. 6c).

**Purkinje cells show no changes of intrinsic properties in *Pogz*-deficient mice.** The reduction in CS frequency represents a change in firing frequency of olivary neurons[36]. However, a reduction in SS firing frequency can be either due to a change in the electrical properties of the PCs or a change in their excitatory and inhibitory inputs. Therefore, we performed whole-cell current-clamp recordings to investigate PC's neuronal excitability. First, we examined the shape and threshold of the action potential (AP, Fig. 6d), using a depolarizing current step. There was no significant difference in shape of AP or its threshold ($P = 0.38$; Control: $−38.5 \pm 6.0$ mV; cKO$^{+/−}$: $−40.3 \pm 6.6$ mV; cKO$^{−/−}$: $−37.4 \pm 6.8$ mV) (Fig. 6e). Second, we measured PC input resistance using a short, low intensity, negative current injection ($−50$ pA of 100 ms; see 'Methods'). There was also no significant difference in PC input resistance ($P = 0.53$; Control: $70.8 \pm 17.8$ MΩ; cKO$^{+/−}$: $76.6 \pm 24.2$ MΩ; cKO$^{−/−}$: $74.4 \pm 20.1$ MΩ) (Fig. 6f). Next, we examined the responses to prolonged current injections (Fig. 6g) to extract the current−frequency curve ($I−f$) (Fig. 6h). All genotypes showed a similar $I−f$ curve with no significant differences ($P = 0.90$). Finally, we examined the properties of the hyperpolarization-activated cationic current $I_h$, which is known to play a major role in the electrophysiological properties of PCs with high relevance to the performance of motor learning behaviors[37–39] and has been suggested as a target of another autism-associated gene, SHANK3[40]. We found no differences between genotypes in either the voltage dependence of the $I_h$ current ($P = 0.79$) (Supplementary Fig. S6a, b) or its kinetics (Fast $\tau$: $P = 0.53$; control: $139.9 \pm 33.9$ ms; cKO$^{+/−}$: $124.6 \pm 36.3$ ms; cKO$^{−/−}$: $150.8 \pm 54.0$ ms; Slow $\tau$: $P = 0.074$; control: $1.02 \pm 0.22$ s; cKO$^{+/−}$: $0.88 \pm 0.30$ s; cKO$^{−/−}$: $0.77 \pm 0.17$ s) (Supplementary Fig. S6c, d). These findings suggest that the observed reduction in SS firing frequency cannot be attributed to changes in PCs excitability and it is likely to represent a change in their synaptic inputs.

**An increase in the amplitude of the inhibitory synaptic input can account for the reduction in Purkinje cells simple spike activity.** PC's SS activity is influenced, to a large extent, by spontaneous excitatory and inhibitory postsynaptic currents (sEPSCs and sIPSCs, respectively). While sEPSCs represent parallel fiber activity, sIPSCs originate at the basket and stellate cells of the molecular layer[41]. Therefore, we examined the frequency and amplitude of sEPSCs (Fig. 7a–d) and sIPSCs (Fig. 7e–h) under conditions that enable separating between these currents (see 'Methods'). Representative examples of sEPSCs recorded from PCs of all genotypes are shown in Fig. 7a. The cumulative probability distributions of the inter-event intervals (IEI) (Fig. 7b) and amplitude (Fig. 7c) did not reveal any differences between the genotypes (Frequency: $P = 0.63$; Control: $1.52 \pm 0.88$ Hz; cKO$^{+/−}$: $1.79 \pm 0.98$ Hz; cKO$^{−/−}$: $1.38 \pm 1.15$ Hz, Amplitude: $P = 0.17$; Control: $10.89 \pm 2.21$ pA; cKO$^{+/−}$: $10.25 \pm 3.03$ pA; cKO$^{−/−}$: $12.95 \pm 4.8$ pA). This result is supported by comparing the average response of the sEPSCs (Fig. 7d, top figure) and their normalized presentation (Bottom figure). Next, we examined sIPSCs and representative examples of sIPSCs are shown in Fig. 7e. Whereas the frequency of sIPSCs was similar among all

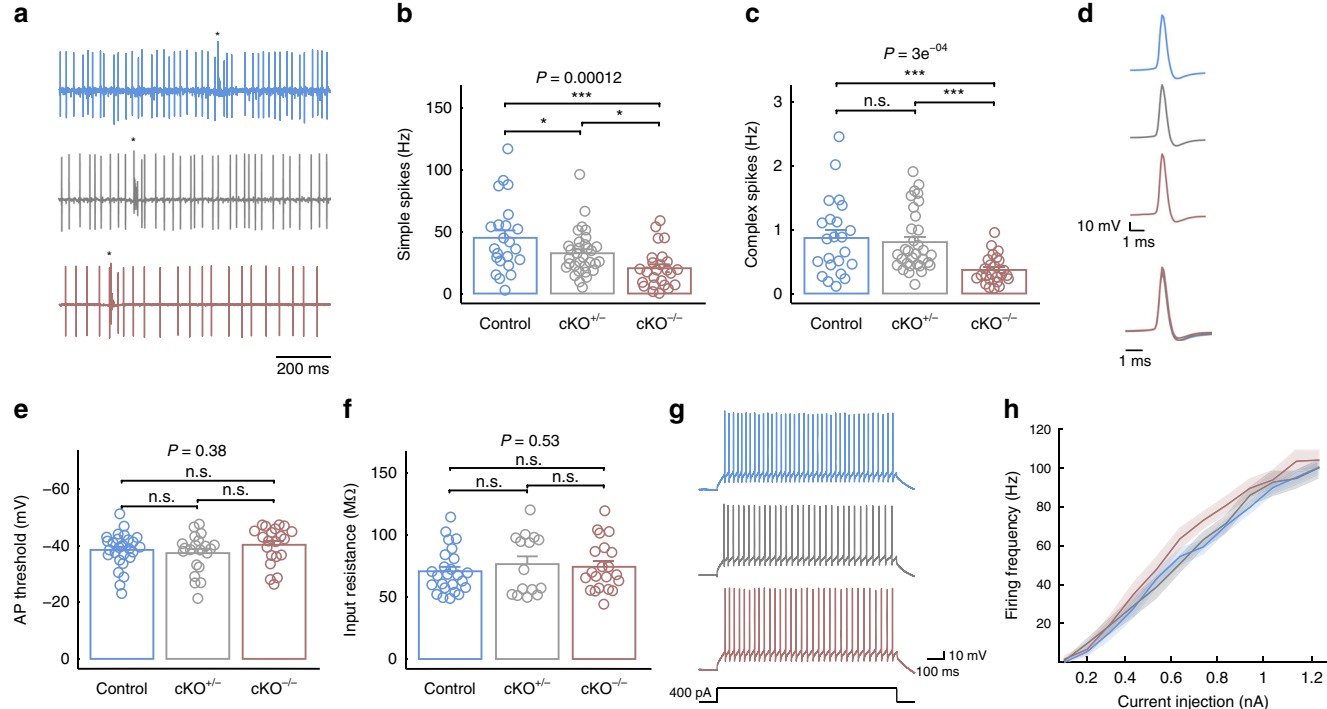

**Fig. 6 *Pogz*-deficient mice show reduced spontaneous firing frequency of Purkinje cells while the PC intrinsic properties are unaffected. a** Single unit activity recorded from PCs in lobules V−VI of the cerebellum. Asterisks indicate complex spikes. **b, c** Spontaneous firing frequency of SS (**b**) and CS (**c**) of all genotypes ($n$ {cells/animals} = 23/5 control (blue), 32/5 cKO$^{+/-}$ (gray), 25/3 cKO$^{-/-}$ (red). **d** AP shape for all genotypes. In the bottom is the superpositioned normalized AP. **e** AP threshold of PCs of all genotypes. $n = 28/3$ control, 21/2 cKO$^{+/-}$, 21/2 cKO$^{-/-}$. **f** Input resistance of PCs. $n = 25/3$ control, 15/2 cKO$^{+/-}$, 21/2 cKO$^{-/-}$. **g** The response of PCs of all genotypes to 400 pA current pulse injections. **h** The average current-to-firing frequency ($I − f$) curve of all genotypes. Shaded area indicates SEM. Repeated-measures ANOVA; $n = 28/3$ control, 21/2 cKO$^{+/-}$, 21/2 cKO$^{-/-}$. The $P$ values at the top of the plots are for association between the quantitative measurements and the number of intact *Pogz* alleles calculated with a linear regression model. Pairwise comparisons between genotypes were calculated by two-tailed $t$ test and the significance is represented by *$P < 0.05$; ***$P < 0.001$; n.s. not significant. Quantitative data are mean ± SEM.

genotypes ($P = 0.5$; Control: $3.56 \pm 3.83$ Hz; cKO$^{+/-}$: $3.21 \pm 3.92$ Hz; cKO$^{-/-}$: $2.8 \pm 3.2$ Hz) (Fig. 7f), we found an increase in the amplitude, which was correlated with *Pogz* deficiency levels ($P = 0.0038$; Control: $20.62 \pm 9.19$ pA; cKO$^{+/-}$ $22.23 \pm 10.02$ pA; cKO$^{-/-}$: $32.61 \pm 18.54$ pA) (Fig. 7g, h, top figure). These differences were not accompanied by a change in the kinetics of the inhibitory events, as the normalized averaged events were identical (Fig. 7h, bottom figure). sIPSCs may represent the firing of inhibitory interneurons or spontaneous vesicular release. To distinguish between these possibilities, we measured miniature synaptic postsynaptic currents (mIPSCs) in the presence of TTX to block the generation of Na$^+$-dependent action potentials. A profound reduction in inhibitory event's frequency (Supplementary Fig. S7) was observed for both control and cKO$^{-/-}$ mice. This strongly supports the notion that the efficacy of the inhibitory synapse is modified rather than spontaneous vesicular release of inhibitory interneurons. Overall, the results imply that the observed reduction in SS activity is likely to represent an increased inhibitory input onto PCs, due to altered synaptic efficacy, while excitatory transmission onto PCs remains intact.

## Discussion

We report the characterization of *Pogz* knockout mouse in the nervous system, which reveals significant findings across genomic, cellular, physiological and behavioral dimensions of neurobiology. Since we studied both heterozygote and homozygote mice for the mutation in *Pogz*, we were able to show that the dosage of *Pogz* is required for repression of transcription in the

brain and is associated with the degree of abnormal behavior and physiology. Our results also suggest that POGZ is required for the proper function of the cerebellum, a brain region that has been consistently linked with ASD[9,42].

The phenotypes that we identified in the *Pogz*-deficient mice resemble several of the characteristics found in human individuals with *POGZ* mutations (White−Sutton syndrome)[15,18,19]. For example, the growth delay and small brain in mice may be related to the short stature and microcephaly seen in patients. How *Pogz* expression in the nervous system regulates the control of body weight is not completely clear. However, the analysis of differentially expressed genes showed enrichment for genes involved in body size, including the *Igf1* and *Igf2* genes. The IGF signaling in the brain is known to be involved in the regulation of body size[43], and may be involved in the growth phenotypes observed in *Pogz*-deficient mice.

Our analysis shows that *Pogz* is involved in the regulation of neurogenesis in the embryonic and adult brain. Neurogenesis was also one of the most significant terms enriched for the differential expressed genes. POGZ was previously found to be essential for normal kinetochore assembly, mitotic chromosome segregation, and for normal mitotic progression[25,44]. Proteins involved in the correct alignment of the mitotic spindle are necessary for the balance between symmetric and asymmetric cell divisions and for controlling the size of the brain[45]. Our study shows that depletion of POGZ results in accelerated exit from the cell cycle in the developing cerebral cortex and a reduction in the number of cells in mitosis, which is consistent with the premature mitotic exit observed when POGZ was knockdown in human cells[25].

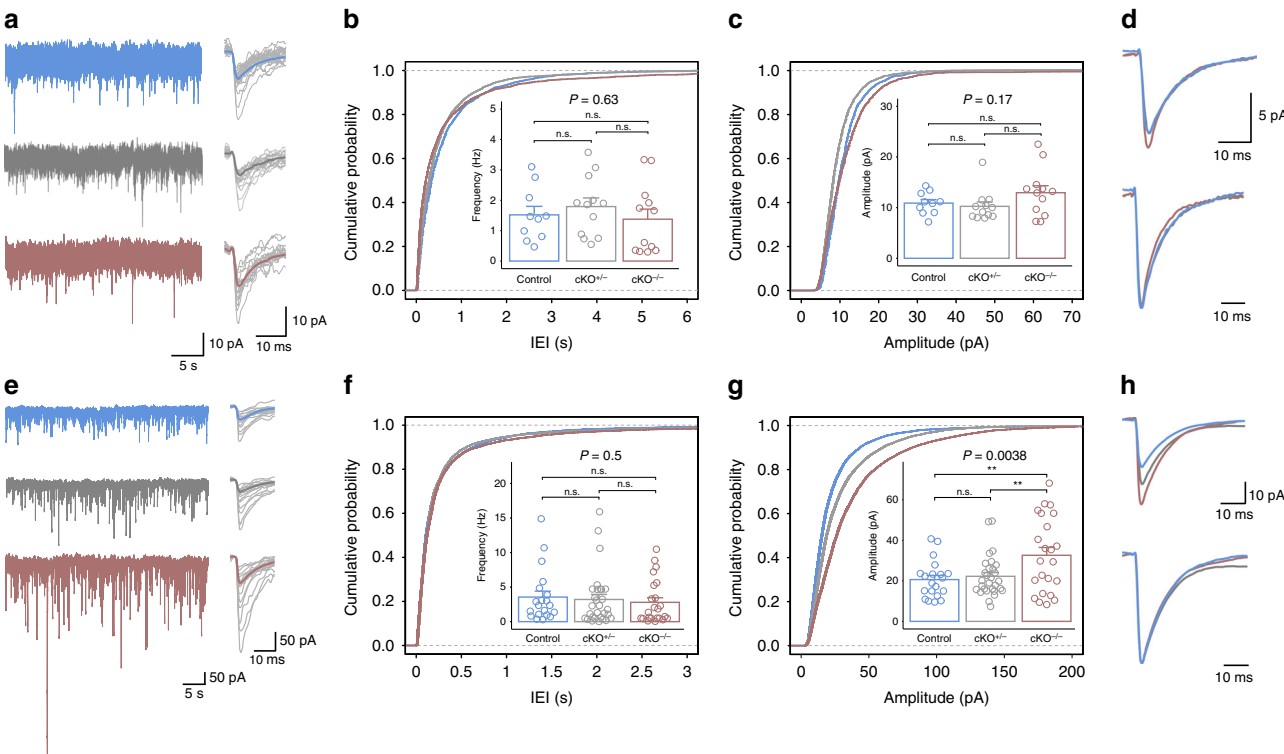

**Fig. 7 Purkinje cells in *Pogz*-deficient mice show an increase in amplitude of inhibitory input, while the excitatory input is unaffected. a** Continuous recordings of sEPSCs from PCs in voltage clamp mode of all genotypes; superpositioned events including the averaged event (in color) are presented on the right. **b** Comparison of the frequency of the sEPSCs (inset, $n = 10/2$ control (blue), 12/2 cKO$^{+/-}$ (gray), 12/2 cKO$^{-/-}$ (red)) and the cumulative probability of the inter-event interval (IEI) of all genotypes ($n$ {IEIs per genotype} = 1814 control, 2569 cKO$^{+/-}$, 2197 cKO$^{-/-}$). **c** Same as (**b**) for the amplitude of the sEPSCs ($n$ {events per genotype} 1824 control, 2581 cKO$^{+/-}$, 2209 cKO$^{-/-}$). **d** Top: The averaged sEPSC of all cells of each genotype. Bottom: Same as in Top after normalization. **e** Continuous recordings of sIPSCs recorded from PCs in voltage clamp mode of all genotypes; superpositioned events including the averaged event (in color) are presented on the right. **f** Comparison of the frequency of sIPSCs (inset, $n = 20/5$ control, 29/5 cKO$^{+/-}$, 23/4 cKO$^{-/-}$) and the cumulative probability of the inter-event interval (IEI) of all genotypes ($n$ {IEIs per genotype} = 8516 control, 11137 cKO$^{+/-}$, 7696 cKO$^{-/-}$). **g** Same as (**f**) for the amplitude of the inhibitory events ($n$ {events per genotype} 8536 control, 11,160 cKO$^{+/-}$, 7719 cKO$^{-/-}$). Note the genotypes differences as manifested in the cumulative probability for the amplitudes of the inhibitory events. **h** Top: The averaged sIPSC of all cells of each genotype. Bottom: Same as in Top after normalization. The $P$ values at the top of the plots are for association between the quantitative measurements and the number of intact *Pogz* alleles calculated with a linear regression model. Pairwise comparisons between genotypes were calculated by two-tailed $t$ test and the significance is represented by: \*\*$P < 0.01$; n.s. not significant. Quantitative data are mean ± SEM.

Acceleration of cell-cycle exit means a decrease in symmetrical division of progenitor cells and increased production of differentiated cells by asymmetric division. The daughter cells that result from asymmetric division can differentiate into neurons or migrate to the subventricular zone to become an intermediate progenitor cell[46], which explains the increased size of Tbr2+ layer in *Pogz*-deficient mice. The decrease in the pool of neuronal precursor cells in the early developmental stage is expected to decrease the final number of neurons. POGZ knockdown was also found to be associated with abnormal amounts of chromosomes[25], which can lead to cell death or genome instability in subsequent division cycles, and may be the cause of the decrease in neurogenesis and cell survival we observed in the dentate gyrus. Reduced adult neurogenesis could lead to impaired hippocampus-dependent learning, including the performance of *Pogz*-deficient mice in the T-maze and water maze[47].

The increased social interactions in *Pogz*-deficient mice may be related to the ASD diagnosis and overly friendly phenotype observed in human individuals with *POGZ* mutations. Other abnormal behaviors included deficits in repetitive digging, working memory and motor coordination and learning. A previous study demonstrated that impaired motor skills in individuals with ASD are associated with de novo mutations across multiple genes[48]. Despite the apparent similarities, one should

note that a complex genotype—phenotype relationship was reported for White—Sutton syndrome as well as for other syndromes associated with ASD, with a large phenotypic heterogeneity even among human individuals that carry similar types of loss-of-function mutations. For example, microcephaly was reported only in a small proportion of cases with White—Sutton syndrome, and a formal ASD diagnosis was reported for roughly half of the individuals[15]. In addition to the general challenge of comparing mouse and human behaviors, and the heterogeneity in phenotypes, our study employs a genetic model that is different from the mutations in humans. Therefore, it should not be expected that a mouse model would recapitulate all the abnormalities seen in patients. Unlike the mutations in humans, our studied mice were based on a conditional model with a deletion in *Pogz* that is specific to the nervous system from a specific development period, which allowed for both heterozygote and homozygote mice to be tested. Across the study, and for many of the measurements (morphological, physiological, expression and behaviors), the heterozygotes (cKO$^{+/-}$) showed an intermediate level between the control and the full KO (cKO$^{-/-}$), which is consistent with an additive effect of *Pogz* on those phenotypes.

We demonstrated that POGZ can repress transcription depending on the HP1-binding zinc finger-like domain, and

consequently, mutations in *Pogz* lead to abnormal expression of multiple genes in the brain. The overlap with genes differentially expressed in GLP and G9a mutant mice supports the involvement of POGZ in a repression complex together with HP1 and GLP/G9a proteins[34]. Based on the enrichment analysis, the changes in expression of genes involved in processes such as metabolism, neurogenesis and synaptic processes can explain the cellular, physiological and behavioral phenotypes of the *Pogz* mouse model, including the decreased body weight, abnormal nervous system electrophysiology and behavior. Some individuals with mutations in *POGZ* were reported to have obesity and skeletal anomalies, which agrees with the enrichment of differentially expressed genes in metabolism and abnormal skeleton morphology. Furthermore, genes involved in neurogenesis and synaptic processes are known to be mutated in ASD, and more generally in NDDs[9,49–51]. In agreement, we found enrichment for the set of ASD genes in genes differentially expressed in *Pogz*-deficient mice.

Multiple lines of evidence point to the cerebellum as a critical brain region affected by *Pogz* deficiency. Not only that mutations in *Pogz* resulted in a profound alteration in gene expression in the cerebellum, we also observed in *Pogz*-deficient mice abnormal motor coordination and learning and decreased firing rates of simple and complex spikes of cerebellar PCs. Although both firing modes are known to participate in motor performance as well as in motor learning[52–60], the mechanism is still debated. Regardless of the exact mechanism of motor performance or motor learning, it is clear that a reduction in simple spike activities in the *Pogz*-deficient mice is bound to affect motor learning.

The reduction in SS activity can be due to alterations in PC intrinsic properties. However, basic properties such as input resistance or current−frequency relationships were unaffected by *Pogz* deficiency. On the other hand, the significant increase in the amplitude of inhibitory inputs is sufficient to account for the recorded reduction in SS activity. It is of special interest that the increase in inhibitory input is manifested as an increase in amplitude of the synaptic current, whereas the frequency of either the inhibitory or the excitatory events were unaffected by the mutation. A straightforward interpretation is that the properties (as for the spontaneous activity) of the local inhibitory interneurons in the cerebellar cortex were unaffected by the mutation. Furthermore, we also show that excitatory transmission to PCs was unaffected. Since the parallel fiber provides excitatory input to both PCs and basket and stellate inhibitory interneurons, one can conclude that the major change that contributes to the reduction in SS activity is the amplitude in the inhibitory input, representing increased synaptic efficacy. It is interesting to note that upregulation of GABA-related genes was found in the gene expression analysis of the cerebellum (e.g. Gabra2 and Gabra3; see Supplementary Table S3), implying a change in $GABA_A$ receptor subunits' composition and/or the number of channels on the membrane.

Reduction in SS firing frequency of PC was previously described in a number of ASD mouse models but this was attributed to changes in PC properties[61,62]. An increase in granule cells sensitivity has also been reported[63], but it cannot account for our observed reduction in SS activity. Finally, enhanced inhibitory input onto the PC, which has been reported in *Shank2* model, can account for the reduction in SS activity[64].

A parallel reduction in CS and SS is unusual given that these two types of PC activity are known to demonstrate a reciprocal relationship. Namely, in any process that involves an increase in CS activity, a reduction in SS was always observed[65–67]. Therefore, a reduction in both types suggests an overall reduction in cerebellar activity. In addition, a reduction in CS activity is bound to reflect a reduction in climbing fiber input. Alterations in synaptic pruning of climbing fibers were previously reported for ASD mouse models, which might affect CS frequency[68,69]. Given the proposed prominent role of climbing fiber in motor learning processes[70], it can account for our observation of reduced motor learning.

In summary, we demonstrated that mutations in *Pogz* results in behavioral deficits and growth impairments that resemble the human condition. We show that POGZ is involved in transcription repression, and expression dysregulation of multiple genes that are involved in neuronal development and function. The dramatic alteration of gene expression in the cerebellum is accompanied by changes in neuronal activity in the cerebellar cortex that can account for the behavioral deficits.

## Methods

**Generation of brain specific *Pogz*-deficient mice.** *Pogz* conditional knockout mice on the background of C57/BL6 were previously generated[27] using the Cre-lox system, with loxP sites flanking exons 13−19. Exons 13−19 include the DDE transposas domain, the CENP DNA binding domain and a part of the zinc fingers domain that binds HP1 proteins (HPZ). To create a mutation that is restricted to the brain, we crossed a conditional *Pogz* flox/flox (*Pogz*[fl/fl]) mice with transgenic Nestin[CRE] (Nes[Cre/+]; *Pogz*[fl/+]) to produce heterozygous (Nes[Cre/+]; *Pogz*[fl/+]), homozygous (Nes[Cre/+]; *Pogz*[fl/fl]) and controls (Nes[+/+]; *Pogz*[fl/fl]) littermates. All mice were tail genotyped using KAPA mouse genotyping kit (KAPA Biosystems KK7302. PCR primers used were: 5′-AATTAAAGGCAGACCTAGCAGGTGG AGG-3′ (forward) and 5′-TAGCACCGC AGACTGCTATCTATTCCTG (reverse) for loxP sequence, 5′-GCGGTCTGGCAGTAAAAACTATC-3′ (forward) and 5′-GTGAAACAGCATTGCTGTCACTT-3′ (reverse) for Nestin-Cre. All mouse studies were approved by the Institutional Animal Care and Use Committees at The Hebrew University of Jerusalem.

**Western blot analysis.** Proteins were extracted from cortices of E14.5 embryos for all three genotypes (control, *Pogz* cKO[+/−] and *Pogz* cKO[−/−]) using modified RIPA buffer (50 mM Tris pH 8.0, 150 mM NaCl, 5 mM ethylenediaminetetraacetic acid (EDTA) pH 8.0, 1% Triton ×100, 0.5% sodium deoxycholate, 0.1% SDS) mixed with protease inhibitors (Sigma-Aldrich). Each tissue sample was ruptured and agitated in RIPA for 1 h in 4°. The lysates were cleared by centrifugation (20 min; $13.8 \times g$; 4°). Before loading, Laemmli sample buffer ×5 with 2% β-mercaptoethanol was added to the samples following by boiling at 95 °C for 5 min. Protein samples were loaded onto 8% acrylamide gel, and wet method was used to transfer the proteins to a polyvinylidene fluoride membrane (PVDF) membrane (Immobilon-P, Millipore IPVH00010). Membranes were air-dried for blocking followed by an overnight incubation in primary antibodies. The next day, membranes were washed in phosphate-buffered saline containing 0.1% Tween-20 (PBST) and incubated in secondary antibodies for 1 h at RT. After another three PBST washes, immunoblots were visualized using SuperSignal chemiluminescent HRP substrates according to the manufacturer's instructions (Thermo Scientific). Unprocessed gels are available in the Source Data file.

**Luciferase reporter assay.** HEK293 cells were cultured on 48-well cell culture plates overnight. Cells were then transiently transfected with 0.1 μg UAS-SV40-luc (luciferase reporter) and 0.04 μg pLR-TK (renilla internal control) and either of 0.25 μg pCDNA3.1-GAL4-POGZ-FL (expressing GAL4 fused to full-length human POGZ) or 0.25 μg pCDNA3.1-GAL4-POGZ-H840A (expressing GAL4 fused to human POGZ mutated in the HPZ domain) using TransIT-2020. Relative luciferase activity was measured, in three independent experiments, after 48 h using the Dual-Luciferase Reporter system (Promega) according to the manufacturer's protocols.

**Brain sectioning and staining**

*Adult brain.* 2−4-month-old mice were euthanized using Isoflurane USP (Piramal Critical Care, NDC 60307-110-25). The brain was then removed and fixed overnight in 4% PFA at 4 °C. Brains were then incubated in 30% sucrose prior to sectioning. Forty micrometer brain sections were made using Leica SM2000R Sliding microtome and stored in 2:1:1 PBS/Glycerol/Ethylene Glycol at −20 °C. Mounted slides were incubated in 0.01 M citrate buffer pH = 6 and heated above 95 °C for 15 min. The slides were then washed in PBS and incubated in 3% $H_2O_2$ for 10 min. Primary antibodies were diluted in 0.5% triton-PBS solution and 2% normal serum (host animal varies according to secondary antibody) and incubated overnight in a humid chamber in RT.

*For IHC staining.* On the following day, slides were washed with PBS and incubated with secondary biotinylated antibody diluted in 0.5% triton-PBS and 2% for 1.5 h in a humid chamber in RT. The slides were then washed and incubated with AB Solution (VECTASTAIN Elite ABC Hrp Kit, Vector Laboratories, PK6100) for 1 h.

Next, slides were washed again in PBS and incubated with DAB substrate (SIG-MAFAST™ 3,3′-Diaminobenzidine tablets, Sigma D4293). Finally, the slides were washed again in PBS and covered with DPX mountant for histology (Sigma 4451).

*For IF staining*. On the following day, slides were washed with PBS, and incubated with the appropriate fluorescently labeled secondary antibodies for 1.5 h in a humid chamber in RT. Finally, slides were vertically air-dried and covered with DAPI flouromount-G (Southern Biotech 0100-20). For BrdU staining, mice were injected i.p. with BrdU (Sigma B5002) 21 days prior to brain extraction.

*Embryonic brain*. Brain sections (14-μm-thickness) prepared with a cryostat (Leica CM1900, Leica Microsystems, Wetzlar, Germany) were treated with HistoVT one (Nakarai tesque Inc, Kyoto, Japan) for antigen retrieval for 20 min and cooling for 20 min. Then sections were incubated in PBS with 0.5% Triton X100 and 0.1% bovine serum albumin (BSA) for 60 min. After washing with PBS-T (PBS with 0.1% Triton X100) three times, the sections were incubated with a primary anti-body overnight at 4 °C. Alexa Fluor 488-labeled IgG was used as a secondary antibody. 4′,6-diamidino-2-phenylindole (DAPI) was used for DNA staining. Images were captured using LSM880 confocal laser microscope (ZEISS, Oberkochen, Germany).

## Assay of cell-cycle exit by in utero electroporation

*Plasmid construction*. Mouse (m) Pogz cDNA was purchased from DNAFORM (Kanagawa, Japan) and cloned into pCAG-Myc vector (Addgene Inc., Cambridge, MA, USA). The following target sequences were inserted into pSuper-puro RNAi vector (OligoEngine, Seattle, WA): shPogz#1, GTATTTGGCTTTGTTTAAA (2391−2399), sh-Pogz#2, CACTAATTGCCAACAACAA (263−281). Numbers indicate the positions from translational start sites. For the control experiments, we used pSuper-H1.shLuc designed against luciferase (CGTACGCGGAATACTTCGA)[71]. All constructs were verified by DNA sequencing.

*Cell culture and transfection*. COS7 cells were cultured essentially as described[72]. Cells were transfected by Lipofectamine 2000 (Life Technologies Japan, Tokyo) according to the manufacturer's instruction.

*In utero electroporation*. Pregnant ICR mice were purchased from SLC Japan (Shizuoka, Japan). At E14.5, pregnant mice were deeply anesthetized with a mix-ture of three drugs: medetomidine (0.75 mg/kg), midazolam (4 mg/kg), and butorphanol (5 mg/kg). Then, 1 μl of solution containing pSuper RNAi vectors (1 μg/μl each) together with GFP vector was injected into the lateral ventricle of embryonic mouse brains with a glass micropipette made from a microcapillary tube (GD-1; Narishige, Tokyo, Japan). The embryo in the uterus was placed between the tweezers-type disc electrode (CUY650-5; NEPA Gene, Chiba, Japan). Electronic pulses (50 ms of 35 V) were charged five times at intervals of 950 ms with an electroporater (NEPA21; NEPA Gene, Chiba, Japan). At least five independent brains were electroporated and analyzed for each experiment. Animals were neither excluded nor died during experimentation. The uterine horns were placed back into the abdominal cavity to allow the embryos to continue normal development. At P0, brains were harvested and coronal brain sections (14-μm-thickness) were prepared with a cryostat (Leica CM1900, Leica Microsystems, Wetzlar, Germany) and were treated with HistoVT one (Nakarai tesque Inc, Kyoto, Japan) for antigen retrieval for 20 min and cooling for 20 min. Then, sections were incubated in PBS with 0.5% Triton X100 and 0.1% BSA for 60 min. After washing with PBST (PBS with 0.1% Triton X100) three times, the sections were incubated with a primary antibody overnight at 4 °C. The next day, sections were washed, incubated with secondary antibody and with 4′,6-diamidino-2-phenylindole (DAPI) for DNA staining. Images were captured using LSM880 confocal laser microscope (ZEISS, Oberkochen, Germany).

*EdU (5-ethynil-2′-deoxyuridine) incorporation experiments*. pCAG-histone 2B (H2B)-EGFP vector was electroporated in utero into embryos with pSuper-H1.shLuc (control) or sh-Pogz#1or #2 at E14.5. Twenty-four hours after electro-poration, pregnant mice were given an intraperitoneal injection of EdU at 25 mg/kg body weight. After 24 h, brains were fixed with 4% paraformaldehyde and frozen sections were made. Then, the ratio of EdU/Ki67/GFP-triple-positive cells to EdU/GFP-double-positive ones was determined. At least five independent brains were electroporated and analyzed for each experiment. Animals were neither excluded nor died during experimentation.

## Antibodies

*IHC and IF*. Rabbit anti-POGZ (ab171934 1:500); Mouse anti-NeuN (MAB377 1:400); Mouse anti-CUTL1/CUX1 (ab54583 1:500); Rat anti-CTIP2 (ab18465 1:500); Rabbit anti-TBR1 (Merck AB10554 1:400); Rabbit anti-Ki67 (ab15580 1:1000); Rat anti-BrdU (ab6326 1:250) Mouse anti-DCX (sc271390 1:200); Mouse anti-SOX2 (sc17320 1:1000); Goat anti-NeuroD (sc1804 1:1000); Rabbit anti-PROX1 (Merck AB5475 1:1000); Rabbit anti-PAX6 (Merck AB2237 1:500); Mouse anti-CNPase (ab6319 1:1000); Mouse anti-Calbindin (Merck C9848 1:1000); Rabbit anti-GFAP (ab7260 1:1000); goat anti-rabbit Alexa fluor 568 (Invitrogen A-11011 1:200); goat anti-mouse Alexa fluor 488 (Invitrogen A-11001 1:200). Goat

anti-Rabbit IgG biotinylated (Vector Labs BA-1000 1:250); Rabbit anti-Goat IgG biotinylated (Vector Labs BA-5000 1:250); Rabbit anti-Rat IgG biotinylated (Vector Labs BA-4000 1:250); Horse anti-Mouse IgG biotinylated (Vector Labs BA-2000 1:250).

*Staining of embryonic brain slices*. Anti-POGZ was prepared as described[26]; chicken anti-GFP (Aves Laboratories GFP-1020 1:1000); rabbit anti-GFP (Thermo Scientific RB-1510 1:1000); rabbit anti-Pax6 (BioLegend PRB 278P 1:500); rabbit anti-TBR2 (ab23345 1:250); rabbit anti-phospho-histone H3 (Cell Signaling Technology 9701 1:400) Alexa Fluor® 488-labeled IgG was used as secondary antibody (ab150077 1:200).

*WB primary antibodies*. Rabbit anti-POGZ (ab167408 1:500); rat anti-βTubulin (ab6160 1:10,000); rabbit polyclonal antibodies against Myc-tag (1:300) and a cytoskeletal protein, Sept11 (1:1000), were produced as described[40,71].

*WB secondary antibodies*. Donkey anti-rabbit IgG HRP (ab7083 1:10,000); donkey anti-rat IgG HRP (ab102265 1:10,000).

**Golgi staining and analysis**. Golgi staining was done using the super Golgi kit (Bioenno 003010). Two-month-old mice were euthanized using Isoflurane USP (Piramal Critical Care, NDC 60307-110-25). Their brains were then removed and immersed in 10 ml of impregnation solution for 12 days. The rest of the protocol was done according to the manufacturer's instructions. Dendritic spines in the dentate gyrus were counted at 40 μm distance from the soma and on a 30 μm dendritic length.

**Behavioral assays**. Mice were group-housed in a room with a 12-h-light, 12-h-dark cycle (lights on at 7:00) in 20−22 °C (humidity 55 ± 10%) with access to food and water ad libitum. All tests were conducted during the light cycle. Behavioral assays were performed with male and female mice at 8−16 weeks of age between 9:00 and 18:00, unless indicated otherwise. Each apparatus was cleaned with diluted EtOH solution before testing of each animal to prevent bias due to olfactory cues. Before all assays, mice were habituated to the behavioral testing core facility for at least 3 days, and specifically to the testing room for 1−2 h prior to the experiment. The behavioral assays were conducted blinded to genotypes during testing and analysis.

*Three-chamber social interaction test*. The three-chamber social interaction apparatus is divided into three interconnected compartments (35 × 20 each). Each side contains a wired cylinder for interaction (referred to as cup; 10.5 cm height × 10 cm bottom diameter). Test mouse was habituated to the apparatus for 5 min. Then, an unfamiliar mouse (stranger 1) was randomly placed in the cup in one of the side compartments, while the other cup remained empty. The test mouse was re-introduced to the apparatus and allowed to explore all three chambers for a 10-min session. Subsequently, a novel unfamiliar mouse (stranger 2) was placed in the previously empty cup and the test mouse had another 10 min to explore the apparatus. The test sessions were recorded using EthoVision XT11 tracking system (Noldus, Leesburg, VA). Both sessions were done in a dark room.

*Direct social interaction*. 8−12-week-old mice were habituated to test room for at least 1 h before the experiment. Stranger mice were conspecific age-matched wild-type mice socially naive to the test mice. At least 90 min prior to the experiment, stranger mice were given identifiable markings on the tails using a black marker pen. A pair of test and stranger mice were introduced into a clean transparent cage (35 × 20 × 18) filled with fresh bedding and the session was recorded for 10 min using Noldus Ethovision XT11 software. Manual quantification of social behaviors was performed by a researcher blinded to the genotype of the test mice. Direct social interaction was determined as any sequence or combination of sequences involving close huddling and sniffing (nose-to-nose and anogenital sniffing) initiated by the test mouse. Times for the total direct social interactions, the fre-quency of nose-to-nose sniffing and anogenital sniffing were measured.

*Marble burying test*. Mice were habituated to a novel testing cage (35 × 20 × 18 cm²) containing a 5-cm layer of chipped cedar wood bedding for 10 min. The mice were returned to their home cage while 20 glass marbles were aligned equidistantly 4 × 5 in the testing cage. Mice were then introduced again to the testing cage and given 30 min to explore it. At the end of the session, the number of buried marbles was recorded. A marble was considered buried if more than half of it was covered with bedding. Results were calculated and plotted as the percentage of marbles buried for each genotype.

*T-maze*. Mice (8−12 weeks old) were habituated to test room at least 1 h before the experiment. After habituation, mice were placed at the base of a T-maze (arm length 30 cm) and were given the choice to freely explore either the right or left arm of the maze for ten consecutive trials. A choice was assumed to be made when mice stepped into an arm with all four paws. At that moment, the gate to that arm was closed and the animal was allowed to explore the arm for 5 s. Then, the mouse was

gently placed back at the base of the T-maze for the next trial. When the mouse chose a similar arm at two consecutive trials, this was scored as a repeat in consecutive trials.

*Morris water maze*. After habituation of 1 h to the testing room, mice were put in a 122 cm water-filled round arena. Water temperature was kept at 22−23 °C and skim milk was added to reduce water's transparency. The arena had four different blue-colored shapes on its sides, one big colorful flower picture above the North section and the experimenter always stood 1 m from it to the South direction. In the Northern-East quarter a $10 \times 10\ cm^2$ transparent platform was submerged 1 cm in the water. Each mouse had four trials per day for a learning period of 5 days. Each trial entry point to the arena was as following: South-East, South, West, North-West and the order was semi-randomized per day. The order was similar across mice per day. Each trial ended either after 60 s if the mouse did not find the platform or 5 s after the mouse found the platform. As the trial ended the mouse was allowed to stay on the platform for 15 s. At the sixth day the platform was removed from the arena and the mice were entered only once from South-East. Tracking was done using the EthoVision XT11 tracking system (Noldus, Leesburg, VA).

*Open field*. The Open Field environment, consisting of a white circular plexigel arena (55 cm diameter), was divided into center (28 cm diameter) and periphery (the outer part of the circle). Mice were placed in one of four edges of the arena (N, W, S, and E) randomly, and their free movement was recorded for 5 min using the EthoVision XT11 tracking system (Noldus, Leesburg, VA).

*Elevated plus maze*. The elevated plus maze apparatus consisted of four elevated arms (52 cm from the floor). The arms were arranged in a cross-like disposition, with two opposite arms being enclosed ($30 \times 5 \times 25\ cm^3$) and two being open ($30 \times 5\ cm^2$), having at their intersection a central square platform that gave access to any of the four arms. All floor surfaces were covered with yellow tape to create a high contrast for the tracking system. Mice were placed in the central platform facing an open arm and could explore the apparatus for 5 min. The session was recorded using EthoVision XT11 tracking system (Noldus, Leesburg, VA).

*Social and nonsocial odors habituation/dishabituation*. Prior to testing, experimenters prepared triplicates of cotton swabs soaked with nonsocial odors or with mice urine and cotton swabs soaked with a smell of a dirty cage. The social odor samples were age- and sex-matched to the study mice. The test mouse was habituated to a new, clean testing cage ($35 \times 20 \times 18\ cm^2$) containing a clean odorless cotton swab for 30 min. During the experiment, each odorous swab was lowered to the cage for 2 min and the mouse was free to sniff it with 1-min intervals. In all trials, the time the mouse spent sniffing the swab was recorded using a manual timer.

*Rotarod*. An accelerating Rotarod (Ugo Basile, Cmoerio VA, Italy) was used in this experiment. Five mice were placed simultaneously on the stationary rod (3 cm diameter). Once all mice were stable on the rod, the motor was turned on and the rod rotation was continuously accelerated (from 4 to 40 rpm over 5 min). Each mouse was given three successive trials (30-min interval between trials) for a maximum of 7 min per trial. Results were measured as the time the mouse stayed on the accelerating rod before falling.

*Horizontal bars*. Motor coordination and forelimb strength were tested using two 38-cm-long steel bars, located 49 cm above the bench surface by a support column at each end. Two bar diameters were used for the experiment, 2 and 4 mm. Test mouse was held by the tail and raised to grasp the horizontal bar in the middle with the forepaws only. The criterion point was either a fall from the bar or the time until one forepaw touches the support column. Maximum test time was 30 s. Mice were first tested on the 2-mm bar and then the 4-mm bar after 1−2 min resting period. Performance (time until falling or touching the support column) scoring was as follows: 1−5 s = 1; 6−10 s = 2; 11−20 s = 3; 21−30 s = 4; after 30 s or placing a forepaw on the side column = 5. Final score was calculated as the sum of scores from both bars tested. This protocol was previously described by Deacon[73].

**Gene expression analysis**. Adult mice hippocampi and cerebella (three males and three females from each genotype) were dissected and RNA was extracted using Qiagen RNeasy lipid tissue mini kit (Qiagen 74804). The RNA integrity was measured using the Agilent BioAnalyzer machine (RNA integrity number (RIN) was above 8.5). Libraries of mRNA for sequencing were made by pulldown of poly (A) RNA using Truseq RNA sample preparation kit (illumina). Library quantity and pooling were measured by Qubit (dsDNS HS) and quality was measured by tape station (HS). Sequencing was done using Next seq 500 high output kit V2 75 cycles (Illumina) on Illumina's NextSeq 500 system with 43 bp paired-end reads (resulting in 50−70 million reads per sample). Denaturing and loading of samples were done according to the manufacturer's instructions.

Reads from RNA-seq were aligned to the mouse genome (mm9), and aligned reads were counted at the gene level using STAR aligner (v250a). Reads that mapped to exons 13−19 at the *Pogz* gene were used to verify the genotypes. Raw

count data were used for differential expression analysis using edgeR[74]. Genes with counts per million (CPM) above 1 in at least six individual samples were included in the analysis. The differential expression analysis was performed using a generalized linear model that included sex and genotypes. Differences were considered statistically significant for FDR < 0.05 (calculated by edgeR package). Exact binomial test was used to test for the unequal proportion of upregulated and downregulated genes. Enrichment of gene ontology (C5 collection: Gene Ontology gene sets), MGI mammalian phenotype, ASD genes and overlap with G9a/GLP was conducted using a gene set enrichment analysis (the fgsea R package).

**Quantitative PCR**. cDNA was synthesized using the Superscript III First-Strand kit for qPCR (Invitrogen 18080093). Real-time PCR was performed using the SsoAdvanced™ Universal SYBR® Green Supermix (Bio-Rad). Fluorescence was monitored and analyzed in a Bio-Rad C1000 Thermal Cycler with a CFX96 real-time system. All experiments were performed in triplicates and analyzed using the $2^{(-\Delta\Delta CT)}$ method, using GAPDH as the reference gene. Primers: murine GAPDH: 5′-TGTTCCTACCCCCAATGTGT-3′ (forward) and 5′-ATTGTCATAC CAGGAAATGAGCTT-3′ (reverse); murine Dach2: 5′-AGTCATGAAGTCACC CTTGGA-3′ (forward) and 5′-TTGGTCAACAGAGTCTCCACA-3′ (reverse).

**Slice preparation for electrophysiology recordings**. Mice (8−12 weeks old) were anesthetized with pentobarbitone (60 mg/kg) and the brain was removed. The cerebellum was rapidly cut and placed in ice-cold physiological cutting solution containing the following (in mM): 124 NaCl, 2.4 KCl, 1 $MgCl_2$, 1.3 $NaH_2PO_4$, 26 $NaHCO_3$, 10 glucose, 2 $CaCl_2$, saturated with 95% $O_2$/5% $CO_2$, pH 7.4 at room temperature. Parasagittal slices (250 μm) of the vermal and hemispherical area were cut using a microslicer (7000 SMZ, Campden Instruments, UK). The slices were incubated with oxygenated physiological solution and maintained at 34 °C. After 1-h incubation, the slices were transferred to a recording chamber and maintained at room temperature under continuous perfusion with the oxygenated physiological solution.

**Whole-cell recordings**. Neurons were visualized using differential interference contrast, infrared microscopy (BX61WI, Olympus, Tokyo, Japan). Purkinje cells were identified by the location of their somata between the granular and molecular layers, by soma size and by the presence of a clear primary dendrite. All recordings were performed at room temperature. For current-clamp experiments, bath solution was the same as the cutting solution. Borosilicate pipettes (4−8 MΩ, pulled on a Narishige pp-83 puller) were filled with internal solution containing the following (in mM): 145 potassium-gluconate, 2.8 KCl, 2 NaCl, 0.1 $CaCl_2$, 2.4 Mg-ATP, 0.4 GTP, 10 HEPES and 1 ethylene glycol-bis(β-aminoethyl ether)-N,N,N′,N′-tetraacetic acid (EGTA) (pH adjusted to 7.3 with KOH). Signals were recorded at 10 kHz and low-pass filtered at 3 kHz. Electrode access resistance was routinely checked, and recordings with values larger than 20 MΩ were not included in the analysis. Experiments were discarded if more than 20% change in holding current was needed to maintain the resting potential. AP threshold was calculated from phase plot analysis where the derivative ($dV/dt$) of a single AP plotted is plotted versus membrane voltage. The point in the phase plot where the increase in $dV/dt$ was larger than 10 mV/ms was defined as the AP threshold. Input resistance was calculated by fitting the voltage response during the short current injection to a double exponential function and the difference between the estimated steady-state value and the initial membrane voltage was divided by the current injection amplitude. For $I-f$ protocol PCs were held at approximately −70 mV, and 1 s current pulse was given every 12 s. In voltage clamp experiments isolating $I_h$ current, bath solution contained the following (in mM): 124 NaCl, 5 KCl, 2 $CaCl_2$, 1.3 $NaH_2PO_4$, 1 $MgCl_2$, 10 glucose, 26 $NaHCO_3$, 1 $NiCl_2$, 0.1 $CdCl_2$, 1 $BaCl_2$, 5 TEACl, 0.0005 TTX, 1 4-aminopyridine (4-AP). Borosilicate pipettes were filled with the same internal solution as in current-clamp experiments. PCs were held at −50 mV, and 4 s voltage pulses were given, every 30 s, from −10 to −40 mV and 2 s pulses from −50 to −70 mV, all in 10-mV steps. Whole-cell capacitive transients and leak currents were not compensated during these recordings. Data were sampled at 10 kHz as in current-clamp experiments, and low-pass filtered at 1 kHz. Voltage clamp recordings were commenced 5−10 min after seal breaking. For 4 s pulses $I_h$ current was defined as the difference between the minimal current recorded immediately after the transient capacitive current and the current measured at the end of the voltage step. As for the 2 s pulses, since the current has not reached steady state, we fitted the current changes during the voltage step by either 1 or 2 exponents, and the estimated steady-state value was subtracted from the minimal current recorded immediately after the transient capacitive current. Analysis of $I_h$ kinetics was done only on traces that were successfully fitted with two exponents, which were most of the traces.

Spontaneous EPSCs (sEPSCs) were pharmacologically isolated using 100 μM Picrotoxin in the bath solution (same as cutting solution). Spontaneous IPSCs (sIPSCs) were pharmacologically isolated using 40 μM DNQX in the bath solution (same as cutting solution) and a cesium-based internal solution containing high $Cl^-$ concentration (in mM): 120 CsCl, 20 TEA-Cl, 4 NaCl, 1 EGTA, 0.1 $CaCl_2$ 10 HEPES, 3 MgATP, 0.4 NaGTP. Both sEPSCs and sIPSCs recordings were commenced at least 5 min after seal breaking, in a holding potential of −70 mV. Series resistance ($R_s$) was routinely checked and recordings were discarded if $R_s$ was

more than 30 MΩ or if there was >15% change in $R_s$. Capacitive transients and leak currents were not compensated during these recordings. Data were sampled at 40 kHz, and low-pass filtered at 1.4 kHz. For analysis, sIPSCs and sEPSCs were detected with a 3.5 pA threshold, and each event was visually inspected for inclusion or rejection. Events could be categorized to fast (<3 ms 10−90% rise time) and slow (>3 ms) rise time populations. For this, all events with at least 20 ms IEI between them were taken in order to avoid cumulative events. In Fig. 7a, e, right, the representative and average events are of fast-rising events of the corresponding cell. In Fig. 7d, h the average events of all cells in each genotype were also generated from the fast-rising event population. Events for the analysis in Fig. 7b, c, f, g were taken from the entire population of events, that is without the 20 ms IEI mentioned above.

A dedicated software based on the LabView platform (National Instruments) was used for monitoring and data acquisition. Experiments were conducted on mice from both sexes for all genotypes. Exceptions were in excitability and $I_h$ experiments, where cKO$^{+/-}$ group consisted of male mice only. All experiments were analyzed using custom scripts in Matlab (Mathworks, Natick, MA, USA).

**In vivo extracellular PC recordings**. Mice were anesthetized with an i.p. injection of a mixture containing ketamine (100 mg/kg) and medetomidine (10 mg/kg). Depth of anesthesia was sufficient to eliminate pinch withdrawal. Anesthesia was maintained throughout the experiment by i.p. Injection of additional supplements of ketamine (25 mg/kg) administered as necessary (typically every 0.5–1 h). Body temperature was kept at 37 °C using a heating blanket. Craniotomy was performed 6.0 mm caudal to Bregma and between midline and 1.5 mm lateral from midline. Recordings were obtained using borosilicate pipettes (3−5 MΩ) pulled on a Narishige pp-83 puller, filled with artificial cerebrospinal fluid (ACSF) containing: NaCl 125 mM, KCl 2.5 mM, NaHCO$_3$ 25 mM, NaH$_2$PO$_4$ 1.25 mM, MgCl$_2$ 1 mM, glucose 25 mM and CaCl$_2$ 2 mM. Experiments were conducted on male mice from all genotypes. In vivo recordings were analyzed using Matlab (Mathworks, Natick, Ma, USA).

**Statistical analysis**. Analyses were carried out with R (http://www.r-project.org). For the conditional mouse model, we used two separate tests. The first test is based on a linear model with additive genetic effects, sex effect as a confounding variable and an interaction term of genotype by sex. The additive effect is a linear effect proportional to the number of *Pogz* intact alleles. Specifically, we refer to the genotypes (cKO$^{-/-}$, cKO$^{+/-}$, control [cKO$^{+/+}$]) → (0, 1, 2) as additive effect. In addition, pairwise two-tailed $t$ test was used to test for the differences between genotypes. For comparisons across time points, a model that included repeated-measures was used (e.g. rotarod, water maze and olfactory assays). To compare between genetic models, we used the AIC function in R. In the dominant model, the cKO$^{+/-}$ and cKO$^{-/-}$ were treated as a single category, in the recessive model, the cKO$^{+/-}$ and control are treated as a single category, and in the additive model the three genotypes are treated as 0, 1, and 2.

**Reporting summary**. Further information on research design is available in the Nature Research Reporting Summary linked to this article.

## Data availability

Data supporting the findings of this work are available within the paper and its Supplementary Information files. RNA sequencing data were deposited to Gene Expression Omnibus (GEO) accession number GSE144648. All other relevant data supporting the findings of this study are available from the authors upon reasonable request. Source data are provided with this paper.

## Code availability

The MATLAB source code for the analysis of whole-cell recordings and in vivo extracellular PC recordings are available upon reasonable request from the authors.

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

## Acknowledgements

This research was supported by the Israel Science Foundation (grant nos. 940/13 and 575/17 to S.S.) and funded in part with Federal funds from the Frederick National Laboratory for Cancer Research, NIH, under Contract HHSN261200800001E (to J.R.K.). The content of this publication does not necessarily reflect the views or policies of the Department of Health and Human Services, nor does mention of trade names, commercial products or organizations imply endorsements by the US Government.

## Author contributions

R.S.-L. and B.T. contributed equally to the work. R.S.-L., B.T., Y.Y., and S.S. designed the experiments, analyzed the data and wrote the manuscript. R.S.-L., B.T., Y.C., M.T. performed and analyzed the behavioral assays. R.S.-L. performed the RNA-seq experiment. B.T. performed and analyzed the electrophysiological experiments. R.S.-L., N.H., K.-i.N., and G.-j.H. performed and analyzed the neurogenesis and cell-cycle experiments. N.T. and G.M.-R. performed the luciferase reporter assay experiment. B.G., K.O.G., and J.R.K. generated the *Pogz* conditional mice. Y.Y. and S.S. supervised the work.

## Competing interests

The authors declare no competing interests.
