## [Peer Review File · Nature Communications]

Reviewers' Comments:

Reviewer #1:

Remarks to the Author:

In this study the authors use a previously generated conditional POGZ knock out mouse to characterize the phenotypes that arise from altering POGZ dosage.

Human genetic studies have identified the POGZ gene as a leading gene of interest in Autism Spectrum Disorder. The present study uses a combination of molecular, behavioral and electrophysiological assays to determine what losing one or both copies of this gene has on mice. Loss of POGZ expression is shown to reduce the body weights and brain size of male and female mice. The reduction in brain size is attributed to reduced neurogenesis based on assays performed in the dentate gyrus. A number of locomotor and social behaviors are shown to be affected in the homozygous knockout animals. POGZ is shown to be able to suppress gene expression in a HP1 dependent fashion. Transcription dysregulation is also shown in both the hippocampus and the cerebellum. Finally, the authors show reductions in Purkinje cell spiking that they attribute to an increase in GABAergic synapse strength.

POGZ is an ASD risk gene of great interest. Thus, work on this mouse in the context of neurodevelopmental disorders is very important. The present study does come off as a bit descriptive. I think that's ok but because the authors have taken this approach I believe the work needs to be more comprehensive. As it stands some work seems unjustifiably limited to specific brain regions or synaptic subclasses. Below I list my major and minor concerns.

Major:

1)The genotype of the control animals is not specified.

2)The authors report that the animals with deficits in POGZ expression exhibit reductions in both body weight and brain weight. The authors go on to suppose that reduced neurogenesis may be responsible for the decrease in brain size. Is it not possible that smaller brain size may simply be the result of a smaller mouse/global growth deficit? Is the cranium of the knockout animal also smaller? It also seems strange and indirect to use adult neurogenesis in the dentate gyrus as a proxy for earlier neurogenesis that ultimately affects brain size. A more direct assay of neurogenesis in a developing brain would be very preferable.

3)Humans that harbor de novo mutations in POGZ that are thought to give rise to ASD-related phenotypes are heterozygous for these mutations. No behavioral phenotypes are observed in cKO+/- mice. Behavioral phenotypes are only seen in cKO-/- . This is not addressed anywhere in the manuscript. The burden is on the authors to provide a compelling argument why the reader should not be concerned about this discrepancy.

4)Only a cursory examination of hippocampal deficits is provided (i.e. granule neuron spine density) despite the observation of significant alteration in ASD and synaptic gene expression here that is not seen in the cerebellum. An electrophysiological examination of synaptic function in CA1 pyramidal neurons in these animals (for example) would make this largely descriptive study more comprehensive and significantly boost my enthusiasm about the work.

5)The authors conclude that because no change in sIPSC frequency is observed there is no effect on glutamatergic neurotransmission in the cerebellum. sIPSC recordings can include both AP-evoked currents and mIPSCs. If a large percentage of the recorded events are mIPSCs rather than AP-evoked current I have very little faith a change in excitatory neurotransmission could be resolved using this indirect method. The authors should directly evaluate glutamatergic synapse function in Purkinje neurons.

Minor:

1) Nestin cre expression is not limited to the brain as stated. It is in both the central and peripheral nervous system.

Reviewer #2:

Remarks to the Author:

This manuscript reports the roles of POGZ in the regulation of brain development and cerebellar functions using a mouse line carrying brain-specific deletion of the *Pogz* gene. This mouse line displays reductions in body weights, brain growth, and neurogenesis. Behaviorally, these mice show strongly suppressed motor function and learning, and enhanced social function, mimicking human symptoms. Mechanistically, they show POGZ proteins repress gene expression through the interaction with HP1 proteins. Consistently, transcriptomic analyses of mouse brain regions (cerebellum and hippocampus) indicate strong increases in gene expression. They also find that cerebellar Purkinje cells show increases in simple and complex spikes that are associated with increased amplitude of the inhibitory synaptic input.

This is a very comprehensive study reporting novel *in vivo* functions of POGZ. Positive aspects of this include the first *in vivo* study exploring POGZ function, defects in brain development and behaviors that mimic human symptoms, the presence of mechanistic explanation (excessive gene expression), and electrophysiology data from the cerebellum. Although the causal relationships between the datasets are not strong, given that POGZ is a very important ASD-risk gene without strong known pathophysiological mechanisms, this would help much the field better understand POGZ-related physiology and pathophysiology.

Major comments

1. People with POGZ mutations show intellectual disability, but the manuscript test only working memory in the T-maze test. Testing spatial and recognition memory using behavioral paradigms such as Morris water maze and novel object recognition (or displaced object recognition) test would help improve the manuscript.

Minor comments

1. The manuscript suggests that *Dach2* upregulation is an important pathophysiological mechanism, but there is no attempt to reverse *Dach2* upregulation and suppress any phenotypes in the mice. Although it may be practically not easy, any related results would help the manuscript.

Reviewer #3:

Remarks to the Author:

This manuscript represents a comprehensive characterization of POGZ mouse conditional KO (both heterozygous and homozygous). There is a strong basis for studying this gene's impact on brain structure and function because likely gene disrupting mutations in POGZ have one of the strongest known associations with autism risk from several *de novo* exome sequencing studies. The conditional knockout (driven by Nestin-Cre) showed a globally reduced body weight, including reduced brain weight. The KO mice additionally showed an increase in social behavior, mirroring observed phenotypes in human patients with likely gene disrupting POGZ mutations. In order to study the gene regulatory function of POGZ, the authors first use a dual luciferase reporter assay showing that POGZ decreases gene expression, but by mutating a site that allows binding with HP1, show that POGZ's negative regulation of gene expression is dependent on HP1. Then the authors ran an RNA-seq experiment and found that, consistent with the finding that POGZ is a

negative regulator of gene expression, reducing the number of POGZ copies generally increases gene expression in the hippocampus and cerebellum. Finally, the authors performed electrophysiological measurements in the cerebellum and provide evidence that reduced spontaneous firing frequency in Purkinje cells is due to the efficacy of the inhibitory input. Overall, I think this is a very valuable study that characterizes the effects of mutations in a gene with strong autism associations on brain structure and function. I have a few small comments that I hope are useful to the authors in improving their manuscript.

Given that the conditional knockout was specific to brain (driven by Nestin-Cre), can the authors comment why they see such a strong impact on body weight?

The focus on adult neurogenesis (Fig 2D-F) is not clear to me given that most (almost all) neurons are born during embryonic development. The results are interesting, but it is not clear to me that they explain the observed microcephaly since so few neurons are derived from adult neural stem cells. Why not study the reason behind the observed microcephaly at the embryonic time period during embryonic neurogenesis? In the discussion the authors state that "it is likely that POGZ also affects developmental neurogenesis", but not experimental evidence for this time period is provided. It would be useful to have a stronger motivation for the adult neurogenesis impacts on brain weight or experimental evidence from the embryonic neurogenesis time period.

It would be useful to label the log₂FC (x-axis) in Figures 5B,E with what the directionality means. Presumably log₂FC > 0 is actually linearly increased expression in cKO^{-/-} vs cKO^{-/+} vs WT?

Supp Fig 1B, the figure legend says SOX2 marks pluripotent stem cells but this is likely not true. These would be adult neural stem cells which are multipotent.

Reviewer #1 (Remarks to the Author):

In this study the authors use a previously generated conditional POGZ knock out mouse to characterize the phenotypes that arise from altering POGZ dosage.

Human genetic studies have identified the POGZ gene as a leading gene of interest in Autism Spectrum Disorder. The present study uses a combination of molecular, behavioral and electrophysiological assays to determine what losing one or both copies of this gene has on mice. Loss of POGZ expression is shown to reduce the body weights and brain size of male and female mice. The reduction in brain size is attributed to reduced neurogenesis based on assays performed in the dentate gyrus. A number of locomotor and social behaviors are shown to be affected in the homozygous knockout animals. POGZ is shown to be able to suppress gene expression in a HP1 dependent fashion. Transcription dysregulation is also shown in both the hippocampus and the cerebellum. Finally, the authors show reductions in Purkinje cell spiking that they attribute to an increase in GABAergic synapse strength.

POGZ is an ASD risk gene of great interest. Thus, work on this mouse in the context of neurodevelopmental disorders is very important. The present study does come off as a bit descriptive. I think that's ok but because the authors have taken this approach I believe the work needs to be more comprehensive. As it stands some work seems unjustifiably limited to specific brain regions or synaptic subclasses. Below I list my major and minor concerns.

Many thanks for acknowledging the importance of this topic and for the extremely valuable suggestions to improve our study (addressed below).

Major:

1)The genotype of the control animals is not specified.

Response: This is now clearly stated in the methods section.

2)The authors report that the animals with deficits in POGZ expression exhibit reductions in both body weight and brain weight. The authors go on to suppose that reduced neurogenesis may be responsible for the decrease in brain size. Is it not possible that smaller brain size may simply be the result of a smaller mouse/global growth deficit? Is the cranium of the knockout animal also smaller? It also seems strange and indirect to use adult neurogenesis in the dentate gyrus as a proxy for earlier neurogenesis that ultimately affects brain size. A more direct assay of neurogenesis in a developing brain would be very preferable.

Response: We agree with this important comment, and indeed brain weight relative to body weight was significantly higher in *Pogz* cKO^{-/-} mice ($P = 0.0062$) (**Figure S2F**). Following this suggestion, we examined neurogenesis in the embryonic cerebral cortex using brain slices from our mouse model and using in utero electroporation with shRNA against POGZ. We now show that POGZ deficiency is associated with reduced mitotic cells, accelerated cell cycle exit and increase in the layer of intermediate neural progenitors, which is consistent with the role of POGZ in normal mitotic progression and regulation of neurogenesis. Acceleration of cell cycle exit means decrease in symmetrical division of progenitor cells, decrease in the pool of neuronal

precursor cells in the early developmental stage, and decrease in the final number of neural cells (**Figures 2D-J, S3A-E**).

3) Humans that harbor de novo mutations in POGZ that are thought to give rise to ASD-related phenotypes are heterozygous for these mutations. No behavioral phenotypes are observed in cKO^{+/-} mice. Behavioral phenotypes are only seen in cKO^{-/-}. This is not addressed anywhere in the manuscript. The burden is on the authors to provide a compelling argument why the reader should not be concerned about this discrepancy.

Response: Thank you for this comment. There are several facts that should be considered when comparing findings from our mouse model with the symptoms in humans. First, there is a very high variance in symptoms among human individuals with loss-of-function mutations in *POGZ*. Second, differences between mouse and human development and brain structure can result in different severity of symptoms. Third, the model we used is a conditional model that does not completely match the genetic insult in humans. One obvious difference is the timing of the genetic insult, which unlike in humans is later during the mouse development (depending on the Nestin promoter), which could result in less severe phenotypes. This explanation is now added to the discussion.

Since we studied both homozygous (with no intact copies of *Pogz*; cKO^{-/-}), heterozygous (with one intact copy of *Pogz*; cKO^{+/-}) and controls (with two intact copies of *Pogz*), we used along the study an additive model as our major statistical test for association between the number of *Pogz* intact copies and different traits and measurements. We also directly tested the differences between genotypes. Across many traits and measurements, the heterozygous are in between the homozygous and the control, which is consistent with the additive model. The reviewer pointed out correctly that in the behaviors studied the direct test between cKO^{+/-} and controls was not formally significant. In the revised version of the manuscript we introduce another behavior test for spatial learning (Morris water maze), which does show significant difference between cKO^{+/-} and controls (**Figure 3D**). Furthermore, we evaluated the fit of three possible genetic models, recessive, additive and dominant, using the Akaike information criterion (AIC) for behaviors that do not include repeated measures (**Table S2**). In the dominant model the cKO^{+/-} and cKO^{-/-} are treated as a single category, in the recessive model the cKO^{+/-} and control are treated as a single category, and in the additive model the three genotypes are treated as 0, 1, and 2. The result of the analysis shows that none of the 6 behaviors are best explained by the dominant model, but 4 out of 6 are best explained by an additive model. Naturally, in traits that follow the additive model there is higher statistical power to find significant differences between the genotypes at the extreme (cKO^{-/-} and control) especially in behavior traits that include a large component of variance between animals.

4) Only a cursory examination of hippocampal deficits is provided (i.e. granule neuron spine density) despite the observation of significant alteration in ASD and synaptic gene expression here that is not seen in the cerebellum. An electrophysiological examination of synaptic function in CA1 pyramidal neurons in these animals (for example) would make this largely descriptive study more comprehensive and significantly boost my enthusiasm about the work.

Response: In our study, we examined spine density, adult neurogenesis and gene expression in the hippocampus. Since we found very significant correlations between differential expression in

the hippocampus and the cerebellum, but the effects were substantially stronger in the cerebellum (**Figure 5G**), we decided to focus on the cerebellum in the electrophysiology part. We agree with the reviewer that extending the electrophysiological examination to other brain regions would benefit the study. Unfortunately, this was hard to achieve during these difficult times, particularly since our labs do not specialize in the hippocampus.

5)The authors conclude that because no change in sIPSC frequency is observed there is no effect on glutamatergic neurotransmission in the cerebellum. sIPSC recordings can include both AP-evoked currents and mIPSCs. If a large percentage of the recorded events are mIPSCs rather than AP-evoked current I have very little faith a change in excitatory neurotransmission could be resolved using this indirect method. The authors should directly evaluate glutamatergic synapse function in Purkinje neurons.

Response: We agree that one should be cautious in the interpretation of presynaptic firing activity based on the analysis of inhibitory synaptic events. Following this important comment, we conducted two additional sets of experiments. First, we examined the effect of TTX which supposed to unravel spontaneous vesicular release. We found a dramatic reduction in the frequency of inhibitory synaptic events (added to the manuscript as **supplementary Figure 7**), which indeed imply that most of the recorded sIPSCs represent AP in inhibitory interneurons – a result that has been described before (Ovsepian & Friel, 2012). Second, as suggested by this reviewer, we conducted a similar set of experiments to analyze the frequency and amplitudes of spontaneous excitatory synaptic events. We found no significant differences between genotypes for both parameters (**Figure 7A-D**).

Ovsepian, S. V., & Friel, D. D. (2012). Enhanced synaptic inhibition disrupts the efferent code of cerebellar purkinje neurons in leaner Cav2.1 Ca²⁺ channel mutant mice. *Cerebellum*, 11(3), 666–680. <https://doi.org/10.1007/s12311-010-0210-9>

Minor:

1)Nestin cre expression is not limited to the brain as stated. It is in both the central and peripheral nervous system.

Response: This was corrected in the revised text, thank you.

Reviewer #2 (Remarks to the Author):

This manuscript reports the roles of POGZ in the regulation of brain development and cerebellar functions using a mouse line carrying brain-specific deletion of the *Pogz* gene. This mouse line displays reductions in body weights, brain growth, and neurogenesis. Behaviorally, these mice show strongly suppressed motor function and learning, and enhanced social function, mimicking human symptoms. Mechanistically, they show POGZ proteins repress gene expression through the interaction with HP1 proteins. Consistently, transcriptomic analyses of mouse brain regions (cerebellum and hippocampus) indicate strong increases in gene expression. They also find that cerebellar Purkinje cells show increases in simple and complex spikes that are associated with

increased amplitude of the inhibitory synaptic input.

This is a very comprehensive study reporting novel in vivo functions of POGZ. Positive aspects of this include the first in vivo study exploring POGZ function, defects in brain development and behaviors that mimic human symptoms, the presence of mechanistic explanation (excessive gene expression), and electrophysiology data from the cerebellum. Although the causal relationships between the datasets are not strong, given that POGZ is a very important ASD-risk gene without strong known pathophysiological mechanisms, this would help much the field better understand POGZ-related physiology and pathophysiology.

We appreciate the reviewer's recognition of the comprehensive scale of our study, as well as the very helpful comments, which we address below.

Major comments

1. People with POGZ mutations show intellectual disability, but the manuscript test only working memory in the T-maze test. Testing spatial and recognition memory using behavioral paradigms such as Morris water maze and novel object recognition (or displaced object recognition) test would help improve the manuscript.

Response: Following this important comment, we studied the mice with Morris water maze and identified significant differences in performance between the genotypes. This is now reported in the results and in new figures (**Figures 3C-D, Figure S4B**).

Minor comments

1. The manuscript suggests that Dach2 upregulation is an important pathophysiological mechanism, but there is no attempt to reverse Dach2 upregulation and suppress any phenotypes in the mice. Although it may be practically not easy, any related results would help the manuscript.

Response: Dach2 is indeed one of the most significant differentially expressed genes, but it is one of many genes that change in the hippocampus and the cerebellum. We attempted to manipulate Dach2 in the brain, but as the reviewer anticipated this was practically not easy, and so we have no results to add following this process. When planning the construct with Dach2 we found that the Dach2 upregulation was of an uncharacterized short isoform that starts in the middle of exon 8. This unusual short isoform seems to be specific to neurons based on RNA-seq performed with major types of cells isolated from mouse cerebral cortex (this has now been added as a **Supplementary Figure 5H-J**).

Reviewer #3 (Remarks to the Author):

This manuscript represents a comprehensive characterization of POGZ mouse conditional KO (both heterozygous and homozygous). There is a strong basis for studying this gene's impact

on brain structure and function because likely gene disrupting mutations in POGZ have one of the strongest known associations with autism risk from several de novo exome sequencing studies. The conditional knockout (driven by Nestin-Cre) showed a globally reduced body weight, including reduced brain weight. The KO mice additionally showed an increase in social behavior, mirroring observed phenotypes in human patients with likely gene disrupting POGZ mutations. In order to study the gene regulatory function of POGZ, the authors first use a dual luciferase reporter assay showing that POGZ decreases gene expression, but by mutating a site that allows binding with HP1, show that POGZ's negative regulation of gene expression is dependent on HP1. Then the authors ran an RNA-seq experiment and found that, consistent with the finding that POGZ is a negative regulator of gene expression, reducing the number of POGZ copies generally increases gene expression in the hippocampus and cerebellum. Finally, the authors performed electrophysiological measurements in the cerebellum and provide evidence that reduced spontaneous firing frequency in Purkinje cells is due to the efficacy of the inhibitory input. Overall, I think this is a very valuable study that characterizes the effects of mutations in a gene with strong autism associations on brain structure and function. I have a few small comments that I hope are useful to the authors in improving their manuscript.

We thank the reviewer for this positive evaluation of our work and for the important criticisms, which we have addressed as detailed below.

Given that the conditional knockout was specific to brain (driven by Nestin-Cre), can the authors comment why they see such a strong impact on body weight?

Response: The mechanisms that are involved in control of body weight are certainly complex and include brain-body interactions. The brain, particularly the hypothalamus, has a key role in the regulation of metabolism, food intake, and body growth. The analysis of differentially expressed genes showed enrichment for genes involved in body size, including IGF1 and IGF2. The IGF signaling in the brain is known to be involved in regulation of body size (Kappeler, et al.). Following this valuable comment, this explanation has now been added to the discussion.

Kappeler, Laurent, et al. "Brain IGF-1 receptors control mammalian growth and lifespan through a neuroendocrine mechanism." *PLoS Biol* 6.10 (2008): e254.

The focus on adult neurogenesis (Fig 2D-F) is not clear to me given that most (almost all) neurons are born during embryonic development. The results are interesting, but it is not clear to me that they explain the observed microcephaly since so few neurons are derived from adult neural stem cells. Why not study the reason behind the observed microcephaly at the embryonic time period during embryonic neurogenesis? In the discussion the authors state that "it is likely that POGZ also affects developmental neurogenesis", but not experimental evidence for this time period is provided. It would be useful to have a stronger motivation for the adult neurogenesis impacts on brain weight or experimental evidence from the embryonic neurogenesis time period.

Response: Thank you for this this important suggestion. Following this comment, we examined neurogenesis in the embryonic cerebral cortex using brain slices from our mouse model and

using in utero electroporation with shRNA against POGZ. We show that POGZ deficiency is associated with reduced mitotic cells, accelerated cell cycle exit and increase in the layer of intermediate neural progenitors, which is consistent with the role of POGZ in normal mitotic progression and regulation of neurogenesis. Acceleration of cell cycle exit means decrease in symmetrical division of progenitor cells, decrease in the pool of neuronal precursor cells in the early developmental stage, and decrease in the final number of neural cells (new data is added to **Figures 2D-J, S3A-E**).

It would be useful to label the log₂FC (x-axis) in Figures 5B,E with what the directionality means. Presumably log₂FC > 0 is actually linearly increased expression in cKO^{-/-} vs cKO^{-/+} vs WT?

Response: Yes, Log₂FC > 0 indeed means genes upregulated in the *Pogz*-deficient mice. We added labels in the figures according to this comment.

Supp Fig 1B, the figure legend says SOX2 marks pluripotent stem cells but this is likely not true. These would be adult neural stem cells which are multipotent.

Thank you. We corrected this error.

Reviewers' Comments:

Reviewer #1:

Remarks to the Author:

The authors' have adequately addressed the concerns outlined in my review. I now recommend publication.

Reviewer #2:

Remarks to the Author:

The authors have fully addressed my review comments. I do not have additional comments.

Reviewer #3:

Remarks to the Author:

The reviewers have addressed all my concerns.